# A Ranking-based, Balanced Loss Function Unifying Classification and Localisation in Object Detection

**Kemal Oksuz, Baris Can Cam, Emre Akbas**[*]**, Sinan Kalkan**[*]
Dept. of Computer Engineering, Middle East Technical University
Ankara, Turkey
`{kemal.oksuz, can.cam, eakbas, skalkan}@metu.edu.tr`

## Abstract

We propose *average Localisation-Recall-Precision* (aLRP), a unified, bounded, balanced and ranking-based loss function for both classification and localisation tasks in object detection. aLRP extends the Localisation-Recall-Precision (LRP) performance metric (Oksuz et al., 2018) inspired from how Average Precision (AP) Loss extends precision to a ranking-based loss function for classification (Chen et al., 2020). aLRP has the following distinct advantages: (i) aLRP is the first ranking-based loss function for both classification and localisation tasks. (ii) Thanks to using ranking for both tasks, aLRP naturally enforces high-quality localisation for high-precision classification. (iii) aLRP provides provable balance between positives and negatives. (iv) Compared to on average $\sim$6 hyperparameters in the loss functions of state-of-the-art detectors, aLRP Loss has only one hyperparameter, which we did not tune in practice. On the COCO dataset, aLRP Loss improves its ranking-based predecessor, AP Loss, up to around $5$ AP points, achieves $48.9$ AP without test time augmentation and outperforms all one-stage detectors. Code available at: `https://github.com/kemaloksuz/aLRPLoss`.

## 1 Introduction

Object detection requires jointly optimizing a classification objective ($\mathcal{L}_c$) and a localisation objective ($\mathcal{L}_r$) combined conventionally with a balancing hyperparameter ($w_r$) as follows:

$$\mathcal{L} = \mathcal{L}_c + w_r \mathcal{L}_r. \tag{1}$$

Optimizing $\mathcal{L}$ in this manner has three critical drawbacks: (D1) It does not correlate the two tasks, and hence, does not guarantee high-quality localisation for high-precision examples (Fig. 1). (D2) It requires a careful tuning of $w_r$ [8, 26, 33], which is prohibitive since a single training may last on the order of days, and ends up with a sub-optimal constant $w_r$ [4, 11]. (D3) It is adversely impeded by the positive-negative imbalance in $\mathcal{L}_c$ and inlier-outlier imbalance in $\mathcal{L}_r$, thus it requires sampling strategies [13, 14] or specialized loss functions [9, 22], introducing more hyperparameters (Table 1).

A recent solution for D3 is to directly maximize Average Precision (AP) with a loss function called AP Loss [7]. AP Loss is a ranking-based loss function to optimize the ranking of the classification outputs and provides balanced training between positives and negatives.

In this paper, we extend AP Loss to address all three drawbacks (D1-D3) with one, unified loss function called average Localisation Recall Precision (aLRP) Loss. In analogy with the link between precision and AP Loss, we formulate aLRP Loss as the average of LRP values [19] over the positive examples on the Recall-Precision (RP) curve. aLRP has the following benefits: (i) It exploits ranking for both classification and localisation, enforcing high-precision detections to have high-quality

---

[*]Equal contribution for senior authorship.

| Input Anchors | Classifier Output (C) | | *Three Possible Localization Outputs* | | | | | |
|---|---|---|---|---|---|---|---|---|
| | | | Pos. Correlated with C ($R_1$) | | Uncorrelated with C ($R_2$) | | Neg. Correlated with C ($R_3$) | |
| | Score | Rank | IoU | Rank | IoU | Rank | IoU | Rank |
| $a_1$ | 1.00 | 1 | 0.95 | 1 | 0.80 | 2 | 0.50 | 4 |
| $a_2$ | 0.90 | -- | -- | -- | -- | -- | -- | -- |
| $a_3$ | 0.80 | 2 | 0.80 | 2 | 0.65 | 3 | 0.65 | 3 |
| $a_4$ | 0.70 | -- | -- | -- | -- | -- | -- | -- |
| $a_5$ | 0.60 | -- | -- | -- | -- | -- | -- | -- |
| $a_6$ | 0.50 | 3 | 0.65 | 3 | 0.50 | 4 | 0.80 | 2 |
| $a_7$ | 0.40 | -- | -- | -- | -- | -- | -- | -- |
| $a_8$ | 0.30 | -- | -- | -- | -- | -- | -- | -- |
| $a_9$ | 0.20 | -- | -- | -- | -- | -- | -- | -- |
| $a_{10}$ | 0.10 | 4 | 0.50 | 4 | 0.95 | 1 | 0.95 | 1 |

**(a)** 3 possible localization outputs ($R_1$-$R_3$) for the same classifier output (C)
(Orange: Positive anchors, Gray: Negative anchors)

| Detector Output | $AP_{50}$ | $AP_{65}$ | $AP_{80}$ | $AP_{95}$ | **AP** |
|---|---|---|---|---|---|
| (C & $R_1$) | 0.51 | 0.43 | 0.33 | 0.20 | **0.37** |
| (C & $R_2$) | 0.51 | 0.39 | 0.24 | 0.02 | **0.29** |
| (C & $R_3$) | 0.51 | 0.19 | 0.08 | 0.02 | **0.20** |

**(b)** Performance in AP = $(AP_{50}+AP_{65}+AP_{80}+AP_{95})/4$

| Detector Output | $\mathcal{L}_c$ | | $\mathcal{L}_r$ | | *Ours* |
|---|---|---|---|---|---|
| | Cross Entropy | AP Loss | L1 Loss | IoU Loss | aLRP Loss |
| (C & $R_1$) | 0.87 | 0.36 | 0.29 | 0.28 | 0.53 |
| (C & $R_2$) | 0.87 | 0.36 | 0.29 | 0.28 | 0.69 |
| (C & $R_3$) | 0.87 | 0.36 | 0.29 | 0.28 | 0.89 |

**(c)** Comparison of different loss functions
(Red: Improper ordering, Green: Proper ordering)

Figure 1: **aLRP Loss enforces high-precision detections to have high-IoUs, while others do not.** **(a)** Classification and three possible localisation outputs for 10 anchors and the rankings of the positive anchors with respect to (wrt) the scores (for $C$) and IoUs (for $R_1$, $R_2$ and $R_3$). Since the regressor is only trained by positive anchors, "–" is assigned for negative anchors. **(b,c)** Performance and loss assignment comparison of $R_1$, $R_2$ and $R_3$ when combined with $C$. When correlation between the rankings of classifier and regressor outputs decreases, performance degrades up to 17 AP (b). While any combination of $\mathcal{L}_c$ and $\mathcal{L}_r$ cannot distinguish them, aLRP Loss penalizes the outputs accordingly (c). The details of the calculations are presented in the Supp.Mat.

Table 1: State-of-the-art loss functions have several hyperparameters (6.4 on avg.). aLRP Loss has only one for step-function approximation (Sec. 2.1). See the Supp. Mat. for descriptions of the required hyperparameters. FL: Focal Loss, CE: Cross Entropy, SL1: Smooth L1, H: Hinge Loss.

| Method | $\mathcal{L}$ | Number of hyperparameters |
|---|---|---|
| AP Loss [7] | AP Loss+$\alpha$ SL1 | 3 |
| Focal Loss [14] | FL+ $\alpha$ SL1 | 4 |
| FCOS [28] | FL+$\alpha$ IoU+$\beta$ CE | 4 |
| DR Loss [24] | DR Loss+$\alpha$ SL1 | 5 |
| FreeAnchor [33] | $\alpha \log(\max(e^{CE} \times e^{\beta SL1}))+\gamma$ FL | 8 |
| Faster R-CNN [25] | CE+$\alpha$ SL1+$\beta$CE+$\gamma$ SL1 | 9 |
| Center Net [8] | FL+FL+$\alpha$ L2+$\beta$ H+$\gamma$ (SL1+SL1) | 10 |
| Ours | aLRP Loss | 1 |

localisation (Fig. 1). (ii) aLRP has a single hyperparameter (which we did not need to tune) as opposed to $\sim$6 in state-of-the-art loss functions (Table 1). (iii) The network is trained by a single loss function that provides provable balance between positives and negatives.

Our contributions are: **(1)** We develop a generalized framework to optimize non-differentiable ranking-based functions by extending the error-driven optimization of AP Loss. **(2)** We prove that ranking-based loss functions conforming to this generalized form provide a natural balance between positive and negative samples. **(3)** We introduce aLRP Loss (and its gradients) as a special case of this generalized formulation. Replacing AP and SmoothL1 losses by aLRP Loss for training RetinaNet improves the performance by up to $5.4$AP, and our best model reaches $48.9$AP without test time augmentation, outperforming all existing one-stage detectors with significant margin.

## 1.1 Related Work

**Balancing $\mathcal{L}_c$ and $\mathcal{L}_r$** in Eq. (1), an open problem in object detection (OD) [21], bears important challenges: Disposing $w_r$, and correlating $\mathcal{L}_c$ and $\mathcal{L}_r$. *Classification-aware regression loss* [3] links the branches by weighing $\mathcal{L}_r$ of an anchor using its classification score. Following Kendall et al.

[11], *LapNet* [4] tackled the challenge by making $w_r$ a learnable parameter based on homoscedastic uncertainty of the tasks. Other approaches [10, 29] combine the outputs of two branches during non-maximum suppression (NMS) at inference. Unlike these methods, aLRP Loss considers the ranking wrt scores for both branches and addresses the imbalance problem naturally.

**Ranking-based objectives in OD:** An inspiring solution for balancing classes is to optimize a ranking-based objective. However, such objectives are discrete wrt the scores, rendering their direct incorporation challenging. A solution is to use black-box solvers for an interpolated AP loss surface [23], which, however, provided only little gain in performance. AP Loss [7] takes a different approach by using an error-driven update mechanism to calculate gradients (Sec. 2). An alternative, DR Loss [24], employs Hinge Loss to enforce a margin between the scores of the positives and negatives. Despite promising results, these methods are limited to classification and leave localisation as it is. In contrast, we propose a single, balanced, ranking-based loss to train both branches.

## 2 Background

### 2.1 AP Loss and Error-Driven Optimization

AP Loss [7] directly optimizes the following loss for AP with intersection-over-union (IoU) thresholded at 0.50:

$$\mathcal{L}^{\mathrm{AP}} = 1 - \mathrm{AP}_{50} = 1 - \frac{1}{|\mathcal{P}|} \sum_{i \in \mathcal{P}} \mathrm{precision}(i) = 1 - \frac{1}{|\mathcal{P}|} \sum_{i \in \mathcal{P}} \frac{\mathrm{rank}^+(i)}{\mathrm{rank}(i)}, \tag{2}$$

where $\mathcal{P}$ is the set of positives; $\mathrm{rank}^+(i)$ and $\mathrm{rank}(i)$ are respectively the ranking positions of the $i$th sample among positives and all samples. $\mathrm{rank}(i)$ can be easily defined using a step function $H(\cdot)$ applied on the difference between the score of $i$ ($s_i$) and the score of each other sample:

$$\mathrm{rank}(i) = 1 + \sum_{j \in \mathcal{P}, j \neq i} H(x_{ij}) + \sum_{j \in \mathcal{N}} H(x_{ij}), \tag{3}$$

where $x_{ij} = -(s_i - s_j)$ is positive if $s_i < s_j$; $\mathcal{N}$ is the set of negatives; and $H(x) = 1$ if $x \geq 0$ and $H(x) = 0$ otherwise. In practice, $H(\cdot)$ is replaced by $x/2\delta + 0.5$ in the interval $[-\delta, \delta]$ (in aLRP, we use $\delta = 1$ as set by AP Loss [7] empirically; this is the only hyperparameter of aLRP – Table 1). $\mathrm{rank}^+(i)$ can be defined similarly over $j \in \mathcal{P}$. With this notation, $\mathcal{L}^{\mathrm{AP}}$ can be rewritten as follows:

$$\mathcal{L}^{\mathrm{AP}} = \frac{1}{|\mathcal{P}|} \sum_{i \in \mathcal{P}} \sum_{j \in \mathcal{N}} \frac{H(x_{ij})}{\mathrm{rank}(i)} = \frac{1}{|\mathcal{P}|} \sum_{i \in \mathcal{P}} \sum_{j \in \mathcal{N}} L_{ij}^{\mathrm{AP}}, \tag{4}$$

where $L_{ij}^{\mathrm{AP}}$ is called a *primary term* which is zero if $i \notin \mathcal{P}$ or $j \notin \mathcal{N}$ [2].

Note that this system is composed of two parts: (i) The differentiable part up to $x_{ij}$, and (ii) the non-differentiable part that follows $x_{ij}$. Chen et al. proposed that an error-driven update of $x_{ij}$ (inspired from perceptron learning [27]) can be combined with derivatives of the differentiable part. Consider the update in $x_{ij}$ that minimizes $L_{ij}^{\mathrm{AP}}$ (and hence $\mathcal{L}^{\mathrm{AP}}$): $\Delta x_{ij} = L_{ij}^{\mathrm{AP*}} - L_{ij}^{\mathrm{AP}} = 0 - L_{ij}^{\mathrm{AP}} = -L_{ij}^{\mathrm{AP}}$, with the target, $L_{ij}^{\mathrm{AP*}}$, being zero for perfect ranking. Chen et al. showed that the gradient of $L_{ij}^{\mathrm{AP}}$ wrt $x_{ij}$ can be taken as $-\Delta x_{ij}$. With this, the gradient of $\mathcal{L}^{\mathrm{AP}}$ wrt scores can be calculated as follows:

$$\frac{\partial \mathcal{L}^{\mathrm{AP}}}{\partial s_i} = \sum_{j,k} \frac{\partial \mathcal{L}^{\mathrm{AP}}}{\partial x_{jk}} \frac{\partial x_{jk}}{\partial s_i} = -\frac{1}{|\mathcal{P}|} \sum_{j,k} \Delta x_{jk} \frac{\partial x_{jk}}{\partial s_i} = \frac{1}{|\mathcal{P}|} \left( \sum_j \Delta x_{ij} - \sum_j \Delta x_{ji} \right). \tag{5}$$

### 2.2 Localisation-Recall-Precision (LRP) Performance Metric

LRP [19, 20] is a metric that quantifies classification and localisation performances jointly. Given a detection set thresholded at a score ($s$) and their matchings with the ground truths, LRP aims to assign an error value within $[0, 1]$ by considering localisation, recall and precision:

$$\mathrm{LRP}(s) = \frac{1}{N_{FP} + N_{FN} + N_{TP}} \left( N_{FP} + N_{FN} + \sum_{k \in TP} \mathcal{E}_{loc}(k) \right), \tag{6}$$

where $N_{FP}$, $N_{FN}$ and $N_{TP}$ are the number of false positives (FP), false negatives (FN) and true positives (TP); A detection is a TP if $\text{IoU}(k) \geq \tau$ where $\tau = 0.50$ is the conventional TP labeling threshold, and a TP has a localisation error of $\mathcal{E}_{loc}(k) = (1 - \text{IoU}(k))/(1 - \tau)$. The detection performance is, then, $\min_s(\text{LRP}(s))$ on the precision-recall (PR) curve, called optimal LRP (oLRP).

## 3 A Generalisation of Error-Driven Optimization for Ranking-Based Losses

Generalizing the error-driven optimization technique of AP Loss [7] to other ranking-based loss functions is not trivial. In particular, identifying the primary terms is a challenge especially when the loss has components that involve only positive examples, such as the localisation error in aLRP Loss.

Given a ranking-based loss function, $\mathcal{L} = \frac{1}{Z} \sum_{i \in \mathcal{P}} \ell(i)$, defined as a sum over individual losses, $\ell(i)$, at positive examples (e.g., Eq. (2)), with $Z$ as a problem specific normalization constant, our goal is to express $\mathcal{L}$ as a sum of *primary terms* in a more general form than Eq. (4):

**Definition 1.** *The **primary term** $L_{ij}$ concerning examples $i \in \mathcal{P}$ and $j \in \mathcal{N}$ is the loss originating from $i$ and distributed over $j$ via a probability mass function $p(j|i)$. Formally,*

$$L_{ij} = \begin{cases} \ell(i)p(j|i), & \text{for } i \in \mathcal{P}, j \in \mathcal{N} \\ 0, & \text{otherwise.} \end{cases} \tag{7}$$

Then, as desired, we can express $\mathcal{L} = \frac{1}{Z} \sum_{i \in \mathcal{P}} \ell(i)$ in terms of $L_{ij}$:

**Theorem 1.** $\mathcal{L} = \frac{1}{Z} \sum_{i \in \mathcal{P}} \ell(i) = \frac{1}{Z} \sum_{i \in \mathcal{P}} \sum_{j \in \mathcal{N}} L_{ij}$. *See Supp.Mat. for the proof.*

Eq. (7) makes it easier to define primary terms and adds more flexibility on the error distribution: e.g., AP Loss takes $p(j|i) = H(x_{ij})/N_{FP}(i)$, which distributes error uniformly (since it is reduced to $1/N_{FP}(i)$) over $j \in \mathcal{N}$ with $s_j \geq s_i$; though, a skewed $p(j|i)$ can be used to promote harder examples (i.e. larger $x_{ij}$). Here, $N_{FP}(i) = \sum_{j \in \mathcal{N}} H(x_{ij})$ is the number of false positives for $i \in \mathcal{P}$.

Now we can identify the gradients of this generalized definition following Chen et al. (Sec. 2.1): The error-driven update in $x_{ij}$ that would minimize $\mathcal{L}$ is $\Delta x_{ij} = L_{ij}{}^* - L_{ij}$, where $L_{ij}{}^*$ denotes "the primary term when $i$ is ranked properly". Note that $L_{ij}{}^*$, which is set to zero in AP Loss, needs to be carefully defined (see Supp. Mat. for a bad example). With $\Delta x_{ij}$ defined, the gradients can be derived similar to Eq. (5). The steps for obtaining the gradients of $\mathcal{L}$ are summarized in Algorithm 1.

---

**Algorithm 1** Obtaining the gradients of a ranking-based function with error-driven update.

---

**Input:** A ranking-based function $\mathcal{L} = (\ell(i), Z)$, and a probability mass function $p(j|i)$
**Output:** The gradient of $\mathcal{L}$ with respect to model output $\mathbf{s}$

1: $\forall i, j$ find primary term: $L_{ij} = \ell(i)p(j|i)$ if $i \in \mathcal{P}, j \in \mathcal{N}$; otherwise $L_{ij} = 0$ (c.f. Eq. (7)).
2: $\forall i, j$ find target primary term: $L_{ij}{}^* = \ell(i)^* p(j|i)$ ($\ell(i)^*$: the error on $i$ when $i$ is ranked properly.)
3: $\forall i, j$ find error-driven update: $\Delta x_{ij} = L_{ij}{}^* - L_{ij} = (\ell(i)^* - \ell(i)) p(j|i)$.
4: **return** $\frac{1}{Z}(\sum_j \Delta x_{ij} - \sum_j \Delta x_{ji})$ for each $s_i \in \mathbf{s}$ (c.f. Eq. (5)).

---

This optimization provides balanced training for ranking-based losses conforming to Theorem 1:

**Theorem 2.** *Training is balanced between positive and negative examples at each iteration; i.e. the summed gradient magnitudes of positives and negatives are equal (see Supp.Mat. for the proof):*

$$\sum_{i \in \mathcal{P}} \left| \frac{\partial \mathcal{L}}{\partial s_i} \right| = \sum_{i \in \mathcal{N}} \left| \frac{\partial \mathcal{L}}{\partial s_i} \right|. \tag{8}$$

---

**Deriving AP Loss.** Let us derive AP Loss as a case example for this generalized framework: $\ell^{\text{AP}}(i)$ is simply $1 - \text{precision}(i) = N_{FP}(i)/\text{rank}(i)$, and $Z = |\mathcal{P}|$. $p(j|i)$ is assumed to be uniform, i.e. $p(j|i) = H(x_{ij})/N_{FP}(i)$. These give us $L_{ij}^{\text{AP}} = \frac{N_{FP}(i)}{\text{rank}(i)} \frac{H(x_{ij})}{N_{FP}(i)} = \frac{H(x_{ij})}{\text{rank}(i)}$ (c.f. $L_{ij}^{\text{AP}}$ in Eq. (4)). Then, since $L_{ij}^{\text{AP}^*} = 0$, $\Delta x_{ij} = 0 - L_{ij}^{\text{AP}} = -L_{ij}^{\text{AP}}$ in Eq. (5).

**Deriving Normalized Discounted Cumulative Gain Loss [17]**: See Supp.Mat.

---

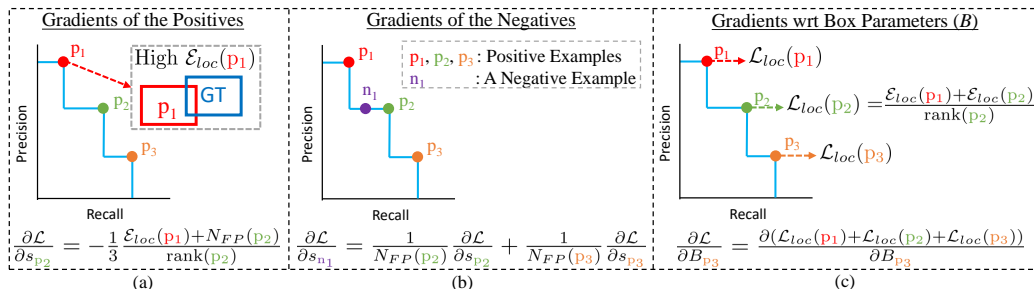

Figure 2: **aLRP Loss assigns gradients to each branch based on the outputs of both branches.** Examples on the PR curve are in sorted order wrt scores ($s$). $\mathcal{L}$ refers to $\mathcal{L}^{\mathrm{aLRP}}$. **(a)** A $p_i$'s gradient wrt its score considers (i) localisation errors of examples with larger $s$ (e.g. high $\mathcal{E}_{loc}(p_1)$ increases the gradient of $s_{p_2}$ to suppress $p_1$), (ii) number of negatives with larger $s$. **(b)** Gradients wrt $s$ of the negatives: The gradient of a $p_i$ is uniformly distributed over the negatives with larger $s$. Summed contributions from all positives determine the gradient of a negative. **(c)** Gradients of the box parameters: While $p_1$ (with highest $s$) is included in total localisation error on each positive, i.e. $\mathcal{L}_{loc}(i) = \frac{1}{\mathrm{rank}(i)}(\mathcal{E}_{loc}(i) + \sum_{k \in \mathcal{P}, k \neq i} \mathcal{E}_{loc}(k)H(x_{ik}))$, $p_3$ is included once with the largest $\mathrm{rank}(p_i)$.

# 4 Average Localisation-Recall-Precision (aLRP) Loss

Similar to the relation between precision and AP Loss, aLRP Loss is defined as the average of LRP values ($\ell^{\mathrm{LRP}}(i)$) of positive examples:

$$\mathcal{L}^{\mathrm{aLRP}} := \frac{1}{|\mathcal{P}|} \sum_{i \in \mathcal{P}} \ell^{\mathrm{LRP}}(i). \tag{9}$$

For LRP, we assume that anchors are dense enough to cover all ground-truths, i.e. $N_{FN} = 0$. Also, since a detection is enforced to follow the label of its anchor during training, TP and FP sets are replaced by the thresholded subsets of $\mathcal{P}$ and $\mathcal{N}$, respectively. This is applied by $H(\cdot)$, and $\mathrm{rank}(i) = N_{TP} + N_{FP}$ from Eq. (6). Then, following the definitions in Sec. 2.1, $\ell^{\mathrm{LRP}}(i)$ is:

$$\ell^{\mathrm{LRP}}(i) = \frac{1}{\mathrm{rank}(i)} \left( N_{FP}(i) + \mathcal{E}_{loc}(i) + \sum_{k \in \mathcal{P}, k \neq i} \mathcal{E}_{loc}(k)H(x_{ik}) \right). \tag{10}$$

Note that Eq. (10) allows using robust forms of IoU-based losses (e.g. generalized IoU (GIoU) [26]) only by replacing IoU Loss (i.e. $1 - \mathrm{IoU}(i)$) in $\mathcal{E}_{loc}(i)$ and normalizing the range to $[0, 1]$.

In order to provide more insight and facilitate gradient derivation, we split Eq. (9) into two as localisation and classification components such that $\mathcal{L}^{\mathrm{aLRP}} = \mathcal{L}_{cls}^{\mathrm{aLRP}} + \mathcal{L}_{loc}^{\mathrm{aLRP}}$, where

$$\mathcal{L}_{cls}^{\mathrm{aLRP}} = \frac{1}{|\mathcal{P}|} \sum_{i \in \mathcal{P}} \frac{N_{FP}(i)}{\mathrm{rank}(i)}, \text{ and } \mathcal{L}_{loc}^{\mathrm{aLRP}} = \frac{1}{|\mathcal{P}|} \sum_{i \in \mathcal{P}} \frac{1}{\mathrm{rank}(i)} \left( \mathcal{E}_{loc}(i) + \sum_{k \in \mathcal{P}, k \neq i} \mathcal{E}_{loc}(k)H(x_{ik}) \right). \tag{11}$$

## 4.1 Optimization of the aLRP Loss

$\mathcal{L}^{\mathrm{aLRP}}$ is differentiable wrt the estimated box parameters, $B$, since $\mathcal{E}_{loc}$ is differentiable [26, 30] (i.e. the derivatives of $\mathcal{L}_{cls}^{\mathrm{aLRP}}$ and $\mathrm{rank}(\cdot)$ wrt $B$ are 0). However, $\mathcal{L}_{cls}^{\mathrm{aLRP}}$ and $\mathcal{L}_{loc}^{\mathrm{aLRP}}$ are not differentiable wrt the classification scores, and therefore, we need the generalized framework from Sec. 3.

Using the same error distribution from AP Loss, the primary terms of aLRP Loss can be defined as $L_{ij}^{\mathrm{aLRP}} = \ell^{\mathrm{LRP}}(i)p(j|i)$. As for the target primary terms, we use the following desired LRP Error:

$$\ell^{\mathrm{LRP}}(i)^* = \frac{1}{\mathrm{rank}(i)} \left( \cancel{N_{FP}(i)}^0 + \mathcal{E}_{loc}(i) + \cancel{\sum_{k \in \mathcal{P}, k \neq i} \mathcal{E}_{loc}(k)H(x_{ik})}^0 \right) = \frac{\mathcal{E}_{loc}(i)}{\mathrm{rank}(i)}, \tag{12}$$

yielding a target primary term, $L_{ij}^{\mathrm{aLRP}*} = \ell^{\mathrm{LRP}}(i)^* p(j|i)$, which includes localisation error and can be non-zero when $s_i < s_j$, unlike AP Loss. Then, the resulting error-driven update for $x_{ij}$ is (line 3 of Algorithm 1):

$$\Delta x_{ij} = \left( \ell^{\mathrm{LRP}}(i)^* - \ell^{\mathrm{LRP}}(i) \right) p(j|i) = -\frac{1}{\mathrm{rank}(i)} \left( N_{FP}(i) + \sum_{k \in \mathcal{P}, k \neq i} \mathcal{E}_{loc}(k) H(x_{ik}) \right) \frac{H(x_{ij})}{N_{FP}(i)}. \tag{13}$$

Finally, $\partial \mathcal{L}^{\mathrm{aLRP}} / \partial s_i$ can be obtained with Eq. (5). Our algorithm to compute the loss and gradients is presented in the Supp.Mat. in detail and has the same time&space complexity with AP Loss.

**Interpretation of the Components:** A distinctive property of aLRP Loss is that classification and localisation errors are handled in a unified manner: i.e. with aLRP, both classification and localisation branches use the entire output of the detector, instead of working in their separate domains as conventionally done. As shown in Fig. 2(a,b), $\mathcal{L}_{cls}^{\mathrm{aLRP}}$ takes into account localisation errors of detections with larger scores ($s$) and promotes the detections with larger IoUs to have higher $s$, or suppresses the detections with high-$s$&low-IoU. Similarly, $\mathcal{L}_{loc}^{\mathrm{aLRP}}$ inherently weighs each positive based on its classification rank (see Supp.Mat. for the weights): the contribution of a positive increases if it has a larger $s$. To illustrate, in Fig. 2(c), while $\mathcal{E}_{loc}(p_1)$ (i.e. with largest $s$) contributes to each $\mathcal{L}_{loc}(i)$; $\mathcal{E}_{loc}(p_3)$ (i.e. with

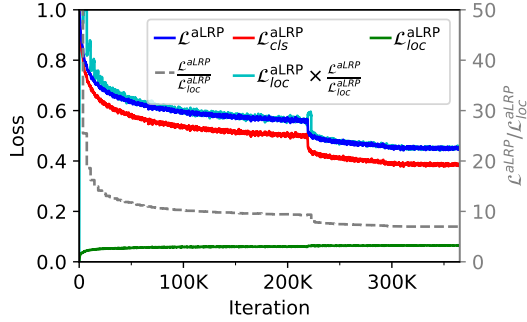

Figure 3: aLRP Loss and its components. The localisation component is self-balanced.

the smallest $s$) only contributes once with a very low weight due to its rank normalizing $\mathcal{L}_{loc}(\mathrm{p}_3)$. Hence, the localisation branch effectively focuses on detections ranked higher wrt $s$.

## 4.2 A Self-Balancing Extension for the Localisation Task

LRP metric yields localisation error only if a detection is classified correctly (Sec. 2.2). Hence, when the classification performance is poor (e.g. especially at the beginning of training), the aLRP Loss is dominated by the classification error ($N_{FP}(i)/\mathrm{rank}(i) \approx 1$ and $\ell^{\mathrm{LRP}}(i) \in [0,1]$ in Eq. (10)). As a result, the localisation head is hardly trained at the beginning (Fig. 3). Moreover, Fig. 3 also shows that $\mathcal{L}_{cls}^{\mathrm{aLRP}}/\mathcal{L}_{loc}^{\mathrm{aLRP}}$ varies significantly throughout training. To alleviate this, we propose a simple and dynamic *self-balancing* (SB) strategy using the gradient magnitudes: note that $\sum_{i \in \mathcal{P}} \left| \partial \mathcal{L}^{\mathrm{aLRP}}/\partial s_i \right| = \sum_{i \in \mathcal{N}} \left| \partial \mathcal{L}^{\mathrm{aLRP}}/\partial s_i \right| \approx \mathcal{L}^{\mathrm{aLRP}}$ (see Theorem 2 and Supp.Mat.). Then, assuming that the gradients wrt scores and boxes are proportional to their contributions to the aLRP Loss, we multiply $\partial \mathcal{L}^{\mathrm{aLRP}}/\partial B$ by the average $\mathcal{L}^{\mathrm{aLRP}}/\mathcal{L}_{loc}^{\mathrm{aLRP}}$ of the previous epoch.

## 5 Experiments

**Dataset:** We train all our models on COCO *trainval35K* set [15] (115K images), test on *minival* set (5k images) and compare with the state-of-the-art (SOTA) on *test-dev* set (20K images).

**Performance Measures:** COCO-style AP [15] and when possible optimal LRP [19] (Sec. 2.2) are used for comparison. For more insight into aLRP Loss, we use Pearson correlation coefficient ($\rho$) to measure correlation between the rankings of classification and localisation, averaged over classes.

**Implementation Details:** For training, we use 4 v100 GPUs. The batch size is 32 for training with $512 \times 512$ images (aLRPLoss500), whereas it is 16 for $800 \times 800$ images (aLRPLoss800). Following AP Loss, our models are trained for 100 epochs using stochastic gradient descent with a momentum factor of 0.9. We use a learning rate of 0.008 for aLRPLoss500 and 0.004 for aLRPLoss800, each decreased by factor 0.1 at epochs 60 and 80. Similar to previous work [7, 8], standard data augmentation methods from SSD [16] are used. At test time, we rescale shorter sides of images to

Table 2: Ablation analysis on COCO *minival*. For optimal LRP (oLRP), lower is better.

| Method | Rank-Based $\mathcal{L}_c$ | Rank-Based $\mathcal{L}_r$ | SB | ATSS | AP | $AP_{50}$ | $AP_{75}$ | $AP_{90}$ | oLRP | $\rho$ |
|---|---|---|---|---|---|---|---|---|---|---|
| AP Loss [7] | ✓ | | | | 35.5 | 58.0 | 37.0 | 9.0 | 71.0 | 0.45 |
| aLRP Loss | ✓ | ✓ (w IoU) | | | 36.9 | 57.7 | 38.4 | 13.9 | 69.9 | 0.49 |
| | ✓ | ✓ (w IoU) | ✓ | | 38.7 | 58.1 | 40.6 | 17.4 | 68.5 | 0.48 |
| | ✓ | ✓ (w GIoU) | ✓ | | 38.9 | 58.5 | 40.5 | 17.4 | 68.4 | 0.48 |
| | ✓ | ✓ (w GIoU) | ✓ | ✓ | 40.2 | 60.3 | 42.3 | 18.1 | 67.3 | 0.48 |

Table 3: SB does not require tuning and slightly outperforms constant weighting for both IoU types.

| $w_r$ | 1 | 2 | 5 | 10 | 15 | 20 | 25 | SB |
|---|---|---|---|---|---|---|---|---|
| w IoU | 36.9 | 37.8 | 38.5 | 38.6 | 38.3 | 37.1 | 36.0 | **38.7** |
| w GIoU | 36.0 | 37.0 | 37.9 | 38.7 | 38.8 | 38.7 | 38.8 | **38.9** |

Table 4: SB is not affected significantly by the initial weight in the first epoch ($w_r$) even for large values.

| $w_r$ | 1 | 50 | 100 | 500 |
|---|---|---|---|---|
| AP | 38.8 | **38.9** | 38.7 | 38.5 |

500 (aLRPLoss500) or 800 (aLRPLoss800) pixels by ensuring that the longer side does not exceed $1.66\times$ of the shorter side. NMS is applied to 1000 top-scoring detections using 0.50 as IoU threshold.

## 5.1 Ablation Study

In this section, in order to provide a fair comparison, we build upon the official implementation of our baseline, AP Loss [5]. Keeping all design choices fixed, otherwise stated, we just replace AP & Smooth L1 losses by aLRP Loss to optimize RetinaNet [14]. We conduct ablation analysis using aLRPLoss500 on ResNet-50 backbone (more ablation experiments are presented in the Supp.Mat.).

**Effect of using ranking for localisation:** Table 2 shows that using a ranking loss for localisation improves AP (from 35.5 to 36.9). For better insight, $AP_{90}$ is also included in Table 2, which shows ~5 points increase despite similar $AP_{50}$ values. This confirms that aLRP Loss does produce high-quality outputs for both branches, and boosts the performance for larger IoUs.

**Effect of Self-Balancing (SB):** Section 4.2 and Fig. 3 discussed how $\mathcal{L}_{cls}^{\text{aLRP}}$ and $\mathcal{L}_{loc}^{\text{aLRP}}$ behave during training and introduced self-balancing to improve training of the localisation branch. Table 2 shows that SB provides +1.8AP gain, similar $AP_{50}$ and +8.4 points in $AP_{90}$ against AP Loss. Comparing SB with constant weighting in Table 3, our SB approach provides slightly better performance than constant weighting, which requires extensive tuning and end up with different $w_r$ constants for IoU and GIoU. Finally, Table 4 presents that initialization of SB (i.e. its value for the first epoch) has a negligible effect on the performance even with very large values. We use 50 for initialization.

**Using GIoU:** Table 2 suggests robust IoU-based regression (GIoU) improves performance slightly.

**Using ATSS:** Finally, we replace the standard IoU-based assignment by ATSS [32], which uses less anchors and decreases training time notably for aLRP Loss: One iteration drops from 0.80s to 0.53s with ATSS (34% more efficient with ATSS) – this time is 0.71s and 0.28s for AP Loss and Focal Loss respectively. With ATSS, we also observe +1.3AP improvement (Table 2). See Supp.Mat. for details.

Hence, we use GIoU [26] as part of aLRP Loss, and employ ATSS [32] when training RetinaNet.

## 5.2 More insight on aLRP Loss

**Potential of Correlating Classification and Localisation.** We analyze two bounds: (i) A *Lower Bound* where localisation provides an inverse ranking compared to classification. (ii) An *Upper Bound* where localisation provides exactly the same ranking as classification. Table 5 shows that correlating ranking can have a significant effect (up to 20 AP) on the performance especially for larger IoUs. Therefore, correlating rankings promises significant

Table 5: Effect of correlating rankings.

| $\mathcal{L}$ | $\rho$ | AP | $AP_{50}$ | $AP_{75}$ | $AP_{90}$ |
|---|---|---|---|---|---|
| aLRP Loss | 0.48 | 38.7 | 58.1 | 40.6 | 17.4 |
| Lower Bound | −1.00 | 28.6 | 58.1 | 23.6 | 5.6 |
| Upper Bound | 1.00 | 48.1 | 58.1 | 51.9 | 33.9 |

improvement (up to $\sim 10$AP). Moreover, while $\rho$ is 0.44 and 0.45 for Focal Loss (results not provided in the table) and AP Loss (Table 2), respectively, aLRP Loss yields higher correlation ($0.48, 0.49$).

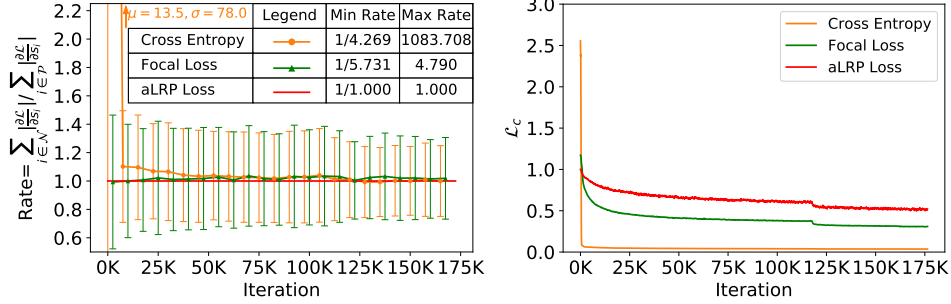

Figure 4: **(left)** The rate of the total gradient magnitudes of negatives to positives. **(right)** Loss values.

**Analysing Balance Between Positives and Negatives.** For this analysis, we compare Cross Entropy Loss (CE), Focal Loss (FL) and aLRP Loss on RetinaNet trained for 12 epochs and average results over 10 runs. Fig. 4 experimentally confirms Theorem 2 for aLRP Loss ($\mathcal{L}_{cls}^{\mathrm{aLRP}}$), as it exhibits perfect balance between the gradients throughout training. However, we see large fluctuations in derivatives of CE and FL (left), which biases training towards positives or negatives alternately across iterations. As expected, imbalance impacts CE more as it quickly drops (right), overfitting in favor of negatives since it is dominated by the error and gradients of these large amount of negatives.

### 5.3 Comparison with State of the Art (SOTA)

Different from the ablation analysis, we find it useful to decrease the learning rate of aLRPLoss500 at epochs 75 and 95. For SOTA comparison, we use the mmdetection framework [6] for efficiency (we reproduced Table 2 using our mmdetection implementation, yielding similar results - see our repository). Table 6 presents the results, which are discussed below:

**Ranking-based Losses.** aLRP Loss yields significant gains over other ranking-based solutions: e.g., compared with AP Loss, aLRP Loss provides +5.4AP for scale 500 and +5.1AP for scale 800. Similarly, for scale 800, aLRP Loss performs 4.7AP better than DR Loss with ResNeXt-101.

**Methods combining branches.** Although a direct comparison is not fair since different conditions are used, we observe a significant margin (around 3-5AP in scale 800) compared to other approaches that combine localisation and classification.

**Comparison on scale 500.** We see that, even with ResNet-101, aLRPLoss500 outperforms all other methods with 500 test scale. With ResNext-101, aLRP Loss outperforms its closest counterpart (HSD) by 2.7AP and also in all sizes ($\mathrm{AP_S}$-$\mathrm{AP_L}$).

**Comparison on scale 800.** For 800 scale, aLRP Loss achieves 45.9 and 47.8AP on ResNet-101 and ResNeXt-101 backbones respectively. Also in this scale, aLRP Loss consistently outperforms its closest counterparts (i.e. FreeAnchor and CenterNet) by 2.9AP and reaches the highest results wrt all performance measures. With DCN [35], aLRP Loss reaches 48.9AP, outperforming ATSS by 1.2AP.

### 5.4 Using aLRP Loss with Different Object Detectors

Here, we use aLRP Loss to train FoveaBox [12] as an anchor-free detector, and Faster R-CNN [25] as a two-stage detector. All models use $500$ scale setting, have a ResNet-50 backbone and follow our mmdetection implementation [6]. Further implementation details are presented in Supp.Mat.

**Results on FoveaBox:** To train FoveaBox, we keep the learning rate same with RetinaNet (i.e. $0.008$) and only replace the loss function by aLRP Loss. Table 7 shows that aLRP Loss outperforms Focal Loss and AP Loss, each combined by Smooth L1 (SL1 in Table 7), by $1.4$ and $3.2$ AP points (and similar oLRP points) respectively. Note that aLRP Loss also simplifies tuning hyperparameters of Focal Loss, which are set in FoveaBox to different values from RetinaNet. One training iteration of Focal Loss, AP Loss and aLRP Loss take $0.34$, $0.47$ and $0.54$ sec respectively.

**Results on Faster R-CNN:** To train Faster R-CNN, we remove sampling, use aLRP Loss to train both stages (i.e. RPN and Fast R-CNN) and reweigh aLRP Loss of RPN by $0.20$. Thus, the number

Table 6: Comparison with the SOTA detectors on COCO *test-dev*. $S$, $\times 1.66$ implies that the image is rescaled such that its longer side cannot exceed $1.66 \times S$ where $S$ is the size of the shorter side. R:ResNet, X:ResNeXt, H:HourglassNet, D:DarkNet, De:DeNet. We use ResNeXt101 64x4d.

| Method | Backbone | Training Size | Test Size | AP | AP$_{50}$ | AP$_{75}$ | AP$_S$ | AP$_M$ | AP$_L$ |
|---|---|---|---|---|---|---|---|---|---|
| *One-Stage Methods* | | | | | | | | | |
| RefineDet [31][‡] | R-101 | $512 \times 512$ | $512 \times 512$ | 36.4 | 57.5 | 39.5 | 16.6 | 39.9 | 51.4 |
| EFGRNet [18][‡] | R-101 | $512 \times 512$ | $512 \times 512$ | 39.0 | 58.8 | 42.3 | 17.8 | 43.6 | 54.5 |
| ExtremeNet [34][*‡] | H-104 | $511 \times 511$ | original | 40.2 | 55.5 | 43.2 | 20.4 | 43.2 | 53.1 |
| RetinaNet [14] | X-101 | $800, \times 1.66$ | $800, \times 1.66$ | 40.8 | 61.1 | 44.1 | 24.1 | 44.2 | 51.2 |
| HSD [2] [‡] | X-101 | $512 \times 512$ | $512 \times 512$ | 41.9 | 61.1 | 46.2 | 21.8 | 46.6 | 57.0 |
| FCOS [28][†] | X-101 | $(640, 800), \times 1.66$ | $800, \times 1.66$ | 44.7 | 64.1 | 48.4 | 27.6 | 47.5 | 55.6 |
| CenterNet [8][*‡] | H-104 | $511 \times 511$ | original | 44.9 | 62.4 | 48.1 | 25.6 | 47.4 | 57.4 |
| ATSS [32][†] | X-101-DCN | $(640, 800), \times 1.66$ | $800, \times 1.66$ | 47.7 | 66.5 | 51.9 | 29.7 | 50.8 | 59.4 |
| *Ranking Losses* | | | | | | | | | |
| AP Loss500 [7][‡] | R-101 | $512 \times 512$ | $500, \times 1.66$ | 37.4 | 58.6 | 40.5 | 17.3 | 40.8 | 51.9 |
| AP Loss800 [7][‡] | R-101 | $800 \times 800$ | $800, \times 1.66$ | 40.8 | 63.7 | 43.7 | 25.4 | 43.9 | 50.6 |
| DR Loss [24][†] | X-101 | $(640, 800), \times 1.66$ | $800, \times 1.66$ | 43.1 | 62.8 | 46.4 | 25.6 | 46.2 | 54.0 |
| *Combining Branches* | | | | | | | | | |
| LapNet [4] | D-53 | $512 \times 512$ | $512 \times 512$ | 37.6 | 55.5 | 40.4 | 17.6 | 40.5 | 49.9 |
| Fitness NMS [29] | De-101 | $512, \times 1.66$ | $768, \times 1.66$ | 39.5 | 58.0 | 42.6 | 18.9 | 43.5 | 54.1 |
| Retina+PISA [3] | R-101 | $800, \times 1.66$ | $800, \times 1.66$ | 40.8 | 60.5 | 44.2 | 23.0 | 44.2 | 51.4 |
| FreeAnchor [33][†] | X-101 | $(640, 800), \times 1.66$ | $800, \times 1.66$ | 44.9 | 64.3 | 48.5 | 26.8 | 48.3 | 55.9 |
| *Ours* | | | | | | | | | |
| aLRP Loss500[‡] | R-50 | $512 \times 512$ | $500, \times 1.66$ | 41.3 | 61.5 | 43.7 | 21.9 | 44.2 | 54.0 |
| aLRP Loss500[‡] | R-101 | $512 \times 512$ | $500, \times 1.66$ | 42.8 | 62.9 | 45.5 | 22.4 | 46.2 | 56.8 |
| aLRP Loss500[‡] | X-101 | $512 \times 512$ | $500, \times 1.66$ | 44.6 | 65.0 | 47.5 | 24.6 | 48.1 | 58.3 |
| aLRP Loss800[‡] | R-101 | $800 \times 800$ | $800, \times 1.66$ | 45.9 | 66.4 | 49.1 | 28.5 | 48.9 | 56.7 |
| aLRP Loss800[‡] | X-101 | $800 \times 800$ | $800, \times 1.66$ | 47.8 | 68.4 | 51.1 | 30.2 | 50.8 | 59.1 |
| aLRP Loss800[‡] | X-101-DCN | $800 \times 800$ | $800, \times 1.66$ | **48.9** | **69.3** | **52.5** | **30.8** | **51.5** | **62.1** |
| *Multi-Scale Test* | | | | | | | | | |
| aLRP Loss800[‡] | X-101-DCN | $800 \times 800$ | $800, \times 1.66$ | 50.2 | 70.3 | 53.9 | 32.0 | 53.1 | 63.0 |

[†]: multiscale training, [‡]: SSD-like augmentation, [*]: Soft NMS [1] and flip augmentation at test time

Table 7: Comparison on FoveaBox [12].

| $\mathcal{L}$ | AP | AP$_{50}$ | AP$_{75}$ | AP$_{90}$ | oLRP |
|---|---|---|---|---|---|
| Focal Loss+SL1 | 38.3 | 57.8 | 40.7 | 15.7 | 68.8 |
| AP Loss+SL1 | 36.5 | 58.3 | 38.2 | 11.3 | 69.8 |
| aLRP Loss (Ours) | **39.7** | **58.8** | **41.5** | **18.2** | **67.2** |

Table 8: Comparison on Faster R-CNN [25]

| $\mathcal{L}$ | AP | AP$_{50}$ | AP$_{75}$ | AP$_{90}$ | oLRP |
|---|---|---|---|---|---|
| Cross Entropy+L1 | 37.8 | 58.1 | 41.0 | 12.2 | 69.3 |
| Cross Entropy+GIoU | 38.2 | 58.2 | 41.3 | 13.7 | 69.0 |
| aLRP Loss (Ours) | **40.7** | **60.7** | **43.3** | **18.0** | **66.7** |

of hyperparameters is reduced from nine (Table 1) to three (two $\delta$s for step function, and a weight for RPN). We validated the learning rate of aLRP Loss as $0.012$, and train baseline Faster R-CNN by both L1 Loss and GIoU Loss for fair comparison. aLRP Loss outperforms these baselines by more than 2.5AP and 2oLRP points while simplifying the training pipeline (Table 8). One training iteration of Cross Entropy Loss (with L1) and aLRP Loss take $0.38$ and $0.85$ sec respectively.

# 6   Conclusion

In this paper, we provided a general framework for the error-driven optimization of ranking-based functions. As a special case of this generalization, we introduced aLRP Loss, a ranking-based, balanced loss function which handles the classification and localisation errors in a unified manner. aLRP Loss has only one hyperparameter which we did not need to tune, as opposed to around 6 in SOTA loss functions. We showed that using aLRP improves its baselines significantly over different detectors by simplifying parameter tuning, and outperforms all one-stage detectors.

## Broader Impact

We anticipate our work to significantly impact the following domains:

1. **Object detection**: Our loss function is unique in many important aspects: It unifies localisation and classification in a single loss function. It uses ranking for both classification and localisation. It provides provable balance between negatives and positives, similar to AP Loss.

   These unique merits will contribute to a paradigm shift in the object detection community towards more capable and sophisticated loss functions such as ours.

2. **Other computer vision problems with multiple objectives**: Problems including multiple objectives (such as instance segmentation, panoptic segmentation – which actually has classification and regression objectives) will benefit significantly from our proposal of using ranking for both classification and localisation.

3. **Problems that can benefit from ranking**: Many vision problems can be easily converted into a ranking problem. They can then exploit our generalized framework to easily define a loss function and to determine the derivatives.

Our paper does not have direct social implications. However, it inherits the following implications of object detectors: Object detectors can be used for surveillance purposes for the betterness of society albeit privacy concerns. When used for detecting targets, an object detector's failure may have severe consequences depending on the application (e.g. self-driving cars). Moreover, such detectors are affected by the bias in data, although they will not try to exploit them for any purposes.

## Acknowledgments and Disclosure of Funding

This work was partially supported by the Scientific and Technological Research Council of Turkey (TÜBİTAK) through a project titled "Object Detection in Videos with Deep Neural Networks" (grant number 117E054). Kemal Öksüz is supported by the TÜBİTAK 2211-A National Scholarship Programme for Ph.D. students. The numerical calculations reported in this paper were performed at TUBITAK ULAKBIM High Performance and Grid Computing Center (TRUBA), and Roketsan Missiles Inc. sources.

## Footnotes

[2] By setting $L_{ij}^{\mathrm{AP}} = 0$ when $i \notin \mathcal{P}$ or $j \notin \mathcal{N}$, we do not require the $y_{ij}$ term used by Chen et al. [7].

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
