[Supplementary Material]

# Supplementary Material for "A Ranking-based, Balanced Loss Function Unifying Classification and Localisation in Object Detection"

**Kemal Oksuz, Baris Can Cam, Emre Akbas**[*]**, Sinan Kalkan**[*]
Dept. of Computer Engineering, Middle East Technical University
Ankara, Turkey
{kemal.oksuz, can.cam, eakbas, skalkan}@metu.edu.tr

## Contents

---

[*]Equal contribution for senior authorship.

Figure S1: Visualization of anchor boxes in the scenarios used in Figure 1. Green and red boxes are positive anchors and ground truths respectively. $p_i$ refers to confidence score of the anchor $a_i$. Note that for all of the scenarios, there are additionally six false positives (see Figure 1), which are excluded in this figure for clarity.

## S1    Details of Figure 1: Comparison of Loss Functions on a Toy Example

This section aims to present the scenario considered in Figure 1 of the main paper. Section S1.1 explains the scenario, Section S1.2 and Section S1.3 clarify how the performance measures (AP, $AP_{50}$, etc.) and loss values (cross-entropy, AP Loss, aLRP Loss, etc.) are calculated.

### S1.1    The Scenario

We assume that the scenario in Figure 1(a) of the paper includes five ground truths of which four of them are detected as true positives with different Intersection-over-Union (IoU) overlaps by three different detectors (i.e. $C\&R_1$, $C\&R_2$, $C\&R_3$). Each detector has a different ranking for these true positives with respect to their IoUs. In addition, the output of each detector contains the same six detections with different scores as false positives. Note that the IoUs of these false positives are marked with "−" in Figure 1(a) since they do not match with any ground truth and therefore their IoUs are not being considered neither by the performance measure (i.e. Average Precision) nor by loss computation.

### S1.2    Performance Evaluation

There are different ways to calculate Average Precision (AP) and loss values. For example, in PASCAL [5] and COCO [9] datasets, the recall domain is divided into 11 and 101 evenly spaced points, respectively, and the precision values at these points are averaged to compute AP for a single IoU threshold.

Here, we present how Average Precision (AP) is calculated in Figure 1(b). Similar to the widely adopted performance metric, COCO-style AP, we use $AP_{IoU}$ with different IoU thresholds. In order to keep things simple but provide the essence of the performance metric, we use four samples with $0.15$ increments (i.e. $\{0.50, 0.65, 0.80, 0.95\}$) instead of ten samples with $0.05$ increments as done by original COCO-style AP.

(a) C&R1 $AP_{50}$  (b) C&R1 $AP_{65}$  (c) C&R1 $AP_{80}$  (d) C&R1 $AP_{95}$

(e) C&R2 $AP_{50}$  (f) C&R2 $AP_{65}$  (g) C&R2 $AP_{80}$  (h) C&R2 $AP_{95}$

(i) C&R3 $AP_{50}$  (j) C&R3 $AP_{65}$  (k) C&R3 $AP_{80}$  (l) C&R3 $AP_{95}$

Figure S2: PR curve of each detector output-$AP_{IoU}$ pair. Rows and columns correspond to different $AP_{IoU}$ and detector outputs respectively. PR curves are interpolated (see the text for more detail).

Table S1: Precision of each detector output-$AP_{IoU}$ pair for evenly spaced recall values. This table is based on the PR curves presented in Fig. S2.

| IoU | Output | Precisions for Different Recalls (R) | | | | | | | | | | $AP_{IoU}$ |
|---|---|---|---|---|---|---|---|---|---|---|---|---|
| | | R=0.1 | R=0.2 | R=0.3 | R=0.4 | R=0.5 | R=0.6 | R=0.7 | R=0.8 | R=0.9 | R=1.0 | |
| 0.50 | $C\&R_1$ | 1.00 | 1.00 | 0.67 | 0.67 | 0.50 | 0.50 | 0.40 | 0.40 | 0.00 | 0.00 | 0.51 |
| | $C\&R_2$ | 1.00 | 1.00 | 0.67 | 0.67 | 0.50 | 0.50 | 0.40 | 0.40 | 0.00 | 0.00 | 0.51 |
| | $C\&R_3$ | 1.00 | 1.00 | 0.67 | 0.67 | 0.50 | 0.50 | 0.40 | 0.40 | 0.00 | 0.00 | 0.51 |
| 0.65 | $C\&R_1$ | 1.00 | 1.00 | 0.67 | 0.67 | 0.50 | 0.50 | 0.00 | 0.00 | 0.00 | 0.00 | 0.43 |
| | $C\&R_2$ | 1.00 | 1.00 | 0.67 | 0.67 | 0.30 | 0.30 | 0.00 | 0.00 | 0.00 | 0.00 | 0.39 |
| | $C\&R_3$ | 0.33 | 0.33 | 0.33 | 0.33 | 0.30 | 0.30 | 0.00 | 0.00 | 0.00 | 0.00 | 0.19 |
| 0.80 | $C\&R_1$ | 1.00 | 1.00 | 0.67 | 0.67 | 0.00 | 0.00 | 0.00 | 0.00 | 0.00 | 0.00 | 0.33 |
| | $C\&R_2$ | 1.00 | 1.00 | 0.20 | 0.20 | 0.00 | 0.00 | 0.00 | 0.00 | 0.00 | 0.00 | 0.24 |
| | $C\&R_3$ | 0.20 | 0.20 | 0.20 | 0.20 | 0.00 | 0.00 | 0.00 | 0.00 | 0.00 | 0.00 | 0.08 |
| 0.95 | $C\&R_1$ | 1.00 | 1.00 | 0.00 | 0.00 | 0.00 | 0.00 | 0.00 | 0.00 | 0.00 | 0.00 | 0.20 |
| | $C\&R_2$ | 0.10 | 0.10 | 0.00 | 0.00 | 0.00 | 0.00 | 0.00 | 0.00 | 0.00 | 0.00 | 0.02 |
| | $C\&R_3$ | 0.10 | 0.10 | 0.00 | 0.00 | 0.00 | 0.00 | 0.00 | 0.00 | 0.00 | 0.00 | 0.02 |

In order to compute a single average precision with an IoU as the conventional true positive labelling threshold, denoted by $AP_{IoU}$, the approaches use different methods for sampling/combining individual precision values on a PR curve. The PR curves corresponding to each detector-$AP_{IoU}$ pair are presented in Figure S2. While drawing these curves, similar to Pascal-VOC and COCO, we also adopt interpolation on the PR curve, which requires keeping the larger precision value in the case that the larger one resides in a lower recall. Then, again similar to what these common methods do for a single AP threshold, we check the precision values on different recall values after splitting the recall axis equally. Here, we use 10 recall points between 0.1 and 1.0 in 0.1 increments. Then, based on the PR curves in Figure S2, we check the precision under these different recall values and present them in Table S1. Having generated these values in Table S1 for each $AP_{IoU}$s, the computation is trivial: Just averaging over these precisions (i.e. row-wise average) yields $AP_{IoU}$s. Finally, averaging over these four $AP_{IoU}$s produces the final detection performance as 0.37, 0.29 and 0.20 for $C\&R_1$, $C\&R_2$, $C\&R_3$ respectively (see Table S1).

## S1.3 Computing the Loss Values

In this section, computing the loss values in Figure 1(c) of the paper is presented in detail. Each section is devoted to a loss function presented in Figure 1(c). To keep things simple, without loss of generality, we make the following assumptions in this section during the calculation of the classification and localisation losses:

1. The classifier has sigmoid non-linearity at the top.

2. There is only one foreground class.

3. Similar to how localisation losses deal with scale- and translation-variance within an image, we assume that each ground truth box is normalized as $[0, 0, 1, 1]$.

4. For each loss, the average of its contributors is reported.

### S1.3.1 Cross-entropy Loss

Cross-entropy Loss of the $i$th example is defined as:

$$\mathcal{L}^{CE}(p_i) = -\mathcal{I}[i \in \mathcal{P}]\log(p_i) - \mathcal{I}[i \in \mathcal{N}]\log(1 - p_i), \tag{S1}$$

such that $p_i$ is the confidence score of the $i$th example obtained by applying the sigmoid activation to the classification logit $s_i$, and $\mathcal{I}[Q]$ is the Iverson bracket which is 1 if the predicate Q is true; or else it is 0.

Seeing that all detector outputs, $C\&R1$, $C\&R2$ and $C\&R3$, involve the same classification output, we apply Eq. (S1) for each anchor on $C$, and then find their average as follows:

$$\mathcal{L}^{CE} = \frac{1}{|\mathcal{P}| + |\mathcal{N}|} \sum_{p_i} \mathcal{L}^{CE}(p_i), \tag{S2}$$

$$= -\frac{1}{10}\Big(\log(1.00) + \log(1 - 0.90) + \log(0.80) + \log(1 - 0.70) + \log(1 - 0.60) + \log(0.50) \tag{S3}$$

$$+ \log(1 - 0.40) + \log(1 - 0.30) + \log(1 - 0.20) + \log(0.10)\Big), \tag{S4}$$

$$= 0.87. \tag{S5}$$

### S1.3.2 Average precision (AP) Loss

The computation of AP Loss is very similar to the $AP_{50}$ computation described in Section S1.2 except that precision is calculated on (and also averaged over) the positive examples instead of the recall values. With this intuition the precision values on the four positives are $1.00, 0.67, 0.50, 0.40$ respectively. Then, AP Loss for the output $C$ in Figure 1 regardless of the localisation output that it is combined with is:

$$\mathcal{L}^{AP} = 1 - AP_{50} = 1 - \frac{1}{|\mathcal{P}|}\sum_{i \in \mathcal{P}} \text{precision}(i),$$

$$= 1 - \frac{1}{4} \times (1.00 + 0.67 + 0.50 + 0.40) = 0.36.$$

### S1.3.3 L1 Loss

For a single ground truth, $\hat{B}_i = [\hat{x}_1, \hat{y}_1, \hat{x}_2, \hat{y}_2]$, and its corresponding detection, $B_i = [x_1, y_1, x_2, y_2]$, L1 Loss is defined simply by averaging over the L1 norm of the differences of the parameters of the detection boxes from their corresponding ground truths:

$$\mathcal{L}^{L1}(\hat{B}_i, B_i) = |\hat{x}_1 - x_1| + |\hat{y}_1 - y_1| + |\hat{x}_2 - x_2| + |\hat{y}_2 - y_2|, \tag{S6}$$

Then, the average L1 Loss is:

$$\mathcal{L}^{L1} = \frac{1}{|\mathcal{P}|} \sum_{i \in \mathcal{P}} \mathcal{L}^{L1}(\hat{B}_i, B_i), \tag{S7}$$

$$= \frac{1}{4} \left( (0.00 + 0.00 + 0.00 + 0.05) + (0.00 + 0.00 + 0.00 + 0.25) \right. \tag{S8}$$

$$+ (0.00 + 0.00 + 0.00 + 0.35) + (0.00 + 0.00 + 0.00 + 0.50) \big), \tag{S9}$$

$$= 0.29. \tag{S10}$$

### S1.3.4 IoU Loss

For a single example, IoU Loss is simply $1 - \mathrm{IoU}(\hat{B}_i, B_i)$. Then, for all three outputs in the scenario (also an instance is illustrated in Figure S1), seeing that the IoU distributions are all equal, the average IoU loss of this detection set is:

$$\mathcal{L}^{\mathrm{IoU}} = \frac{1}{|\mathcal{P}|} \sum_{i \in \mathcal{P}} 1 - \mathrm{IoU}(\hat{B}_i, B_i),$$

$$= \frac{1}{4} \left( (1 - 0.95) + (1 - 0.80) + (1 - 0.65) + (1 - 0.50) \right) = 0.28.$$

### S1.3.5 aLRP Loss

This section calculates aLRP Loss value in the scenario, and therefore we believe that at the same time it is also a toy example to present more insight on aLRP Loss.

First, let us recall the definition of aLRP Loss from the paper to simplify tracking this section. aLRP Loss is defined as:

$$\mathcal{L}^{\mathrm{aLRP}} := \frac{1}{|\mathcal{P}|} \sum_{i \in \mathcal{P}} \ell^{\mathrm{LRP}}(i), \tag{S11}$$

such that

$$\ell^{\mathrm{LRP}}(i) = \frac{1}{\mathrm{rank(i)}} \left( N_{FP}(i) + \mathcal{E}_{loc}(i) + \sum_{k \in \mathcal{P}, k \neq i} \mathcal{E}_{loc}(k) H(x_{ik}) \right), \tag{S12}$$

where $\mathcal{E}_{loc}(k) = (1 - \mathrm{IoU}(k))/(1 - \tau)$. Here, we take $H(x)$ as a step function instead of its approximation for simplicity.

Table S2 presents the computation of aLRP values including all by-products for each of the four positive anchors in $C\&R_1$, $C\&R_2$ and $C\&R_3$. Given the table presented in Figure 1(a) in the paper, we present how each column is derived in the following steps:

1. $1 - \mathrm{IoU}(i)$ is simply the IoU Loss of the positive anchors after prediction.
2. $\mathcal{E}_{loc}(i) = (1 - \mathrm{IoU}(i))/(1 - \tau)$ such that $\tau = 0.5$.
3. Define a cumulative sum: $\mathrm{cumsum}(\mathcal{E}_{\mathrm{loc}})(i) = \mathcal{E}_{loc}(i) + \sum_{k \in \mathcal{P}, k \neq i} \mathcal{E}_{loc}(k) H(x_{ik})$ (see Eq. S12). Note that this simply corresponds to a cumulative sum on a positive example using the examples with larger scores and itself. Accordingly, in Table S2, $\mathrm{cumsum}(\mathcal{E}_{\mathrm{loc}})(i)$ is calculated by summing $\mathcal{E}_{loc})(i)$ column over anchors until (and including) $i$th example.
4. $N_{FP}(i)$ is the number of negative examples with larger scores than the $i$th positive anchor. (See Section 3 for the formal definition.)
5. $\mathrm{rank}(i)$ is the rank of an example within positives and negatives. (See Section 2 for the formal definition.)
6. Then using $\mathrm{cumsum}(\mathcal{E}_{\mathrm{loc}})(i)$, $N_{FP}(i)$ and $\mathrm{rank}(i)$, LRP error on a positive example can be computed as:

$$\ell^{\mathrm{LRP}}(i) = \frac{N_{FP}(i) + \mathrm{cumsum}(\mathcal{E}_{\mathrm{loc}})(i)}{\mathrm{rank}(i)}. \tag{S13}$$

7. In the rightmost column, aLRP Loss of a detector, $\mathcal{L}^{aLRP}$, is determined simply averaging over these single LRP values (i.e. $\ell^{\mathrm{LRP}}(i)$ ) on positives.

Table S2: Per-box calculation of $\mathcal{L}_{aLRP}$

| Output | Anchor | $1 - \text{IoU}(i)$ | $\mathcal{E}_{loc}(i)$ | $\text{cumsum}(\mathcal{E}_{\text{loc}})(i)$ | $N_{FP}(i)$ | rank(i) | $\ell^{\text{LRP}}(i)$ | $\mathcal{L}^{\text{aLRP}}$ |
|---|---|---|---|---|---|---|---|---|
| C&R1 | $a_1$ | 0.05 | 0.10 | 0.10 | 0.00 | 1.00 | 0.10 | |
| | $a_3$ | 0.20 | 0.40 | 0.50 | 1.00 | 3.00 | 0.50 | 0.53 |
| | $a_6$ | 0.35 | 0.70 | 1.20 | 3.00 | 6.00 | 0.70 | |
| | $a_{10}$ | 0.50 | 1.00 | 2.20 | 6.00 | 10.00 | 0.82 | |
| C&R2 | $a_1$ | 0.20 | 0.40 | 0.40 | 0.00 | 1.00 | 0.40 | |
| | $a_3$ | 0.35 | 0.70 | 1.10 | 1.00 | 3.00 | 0.70 | 0.69 |
| | $a_6$ | 0.50 | 1.00 | 2.10 | 3.00 | 6.00 | 0.85 | |
| | $a_{10}$ | 0.20 | 0.40 | 2.50 | 6.00 | 10.00 | 0.82 | |
| C&R3 | $a_1$ | 0.50 | 1.00 | 1.00 | 0.00 | 1.00 | 1.00 | |
| | $a_3$ | 0.35 | 0.70 | 1.70 | 1.00 | 3.00 | 0.90 | 0.89 |
| | $a_6$ | 0.20 | 0.40 | 2.10 | 3.00 | 6.00 | 0.85 | |
| | $a_{10}$ | 0.05 | 0.10 | 2.20 | 6.00 | 10.00 | 0.82 | |

Table S3: Common loss functions and the hyperparameters in their definitions.

| | Loss Function | Type | Number & Usage of the Hyper-parameters | |
|---|---|---|---|---|
| $\mathcal{L}_c$ | Cross-entropy [10, 15] | Score-based | 0 | Sampling methods are required |
| | $\alpha$-bal. Cross-entropy[8] | Score-based | 1 | The weight of the foreground anchors |
| | Focal Loss [8] | Score-based | 2 | The weight of the foreground anchors |
| | | | | Modulating factor for hard examples |
| | AP Loss [2] | Ranking-based | 1 | Smoothness of the step function |
| | DR Loss [14] | Ranking-based | 3 | Regularizer for foreground distribution |
| | | | | Regularizer for background distribution |
| | | | | Smoothness of the loss |
| $\mathcal{L}_r$ | Smooth $L_1$ [6] | $l_p$-based | 1 | Cut-off from $L_1$ loss to $L_2$ loss |
| | Balanced $L_1$ [12] | $l_p$-based | 2 | The weight of the inlier anchors |
| | | | | Upper bound of the loss value |
| | IoU Loss [16] | IoU-based | 0 | - |

## S2    Details of Table 1: Hyperparameters of the Loss Functions and Models

This section presents the hyperparameters of the common loss functions in object detection and how they are combined by different models in Table 1.

### S2.1    Hyperparameters of the Individual Loss Functions

Table S3 presents common loss functions and their hyperparameters. Note that since any change in these hyperparameter change the value of the loss function and affects its contribution to the multi-task learning nature of object detection, and, therefore $w_r$ also needs to be retuned.

### S2.2    Hyperparameters of the Loss Functions of the Models

This section discusses the loss functions of the methods discussed in Table 1 in the paper. Obviously, AP Loss [2], Focal Loss [8] and DR Loss [14] follow the formulation in Equation 1. Hence using Table S3, their total number of hyperparameters is easy to see. For example, DR Loss with three hyper-parameters is combined with Smooth L1, which has one hyperparameter. Including the weight of the localisation component, five hyper-parameters are required to be tuned.

Other architectures in Table 1 use more than two loss functions in order to learn different aspects to improve the performance:

- FCOS [17] includes an additional centerness branch to predict the centerness of the pixels, which is trained by an additional cross entropy loss.
- FreeAnchor [20] aims simultaneously to learn the assignment of the anchors to the ground truths by modeling the loss function based on maximum likelihood estimation. In Table 1,

one can easily identify six hyper-parameters from the loss formulation of the Free Anchor and exploiting Table S3. Moreover, the inputs of the focal loss are subject to a saturated linear function with two hyperparameters, which makes eight in total.

- A different set of approaches, an example of which is Faster R-CNN [15], uses directly cross entropy loss. However, cross entropy loss requires to be accompanied by a sampling method by which a set of positive and negative examples are sampled from the set of labelled anchors to alleviate the significant class imbalance. Even for random sampler, two of the following needs to be tuned in order to ensure stable training: (i) Number of positive examples (ii) Number of negative examples (iii) The rate between positives and negatives. Moreover, for a two-stage detector, these should be tuned for both stages, which brings about additional four hyper-parameters. That's why Faster R-CNN [15] in Table 1 requires nine hyperparameters.

- Finally, CenterNet [4], as a state-of-the-art bottom-up method, has a loss function with several components while learning to predict the centers and the corners. It combines six individual losses, one of which is Hinge Loss with one hyperparameter. Considering the type of each, the loss function of CenterNet [4] has 10 hyper-parameters in total.

## S3 Proofs of Theorem 1 and Theorem 2

This section presents the proofs for the theorems presented in our paper.

**Theorem 1.** $\mathcal{L} = \frac{1}{Z} \sum_{i \in \mathcal{P}} \ell(i) = \frac{1}{Z} \sum_{i \in \mathcal{P}} \sum_{j \in \mathcal{N}} L_{ij}.$

*Proof.* The ranking function is defined as:

$$\mathcal{L} = \frac{1}{Z} \sum_{i \in \mathcal{P}} \ell(i). \tag{S14}$$

Since $\forall i \sum_{j \in \mathcal{N}} p(j|i) = 1$, we can rewrite the definition as follows:

$$\frac{1}{Z} \sum_{i \in \mathcal{P}} \ell(i) \left( \sum_{j \in \mathcal{N}} p(j|i) \right). \tag{S15}$$

Reorganizing the terms concludes the proof as follows:

$$\frac{1}{Z} \sum_{i \in \mathcal{P}} \sum_{j \in \mathcal{N}} \ell(i) p(j|i) = \frac{1}{Z} \sum_{i \in \mathcal{P}} \sum_{j \in \mathcal{N}} L_{ij}. \tag{S16}$$

$\square$

**Theorem 2.** *Training is balanced between positive and negative examples at each iteration; i.e. the summed gradient magnitudes of positives and negatives are equal:*

$$\sum_{i \in \mathcal{P}} \left| \frac{\partial \mathcal{L}}{\partial s_i} \right| = \sum_{i \in \mathcal{N}} \left| \frac{\partial \mathcal{L}}{\partial s_i} \right|. \tag{S17}$$

*Proof.* The gradients of a ranking-based loss function are derived as (see Algorithm 1 and Equation 5 in the paper):

$$\frac{\partial \mathcal{L}}{\partial s_i} = \frac{1}{Z} \left( \sum_j \Delta x_{ij} - \sum_j \Delta x_{ji} \right) = \frac{1}{Z} \sum_j \Delta x_{ij} - \frac{1}{Z} \sum_j \Delta x_{ji}, \tag{S18}$$

such that $\Delta x_{ij}$ is the update for $x_{ij}$s and defined as $\Delta x_{ij} = L_{ij}^* - L_{ij}$. Note that both $L_{ij}$ and $L_{ij}^*$ can be non-zero only if $i \in \mathcal{P}$ and $j \in \mathcal{N}$ following the definition of the primary term. Hence, the same applies to $\Delta x_{ij}$: if $i \notin \mathcal{P}$ or $j \notin \mathcal{N}$, then $\Delta x_{ij} = 0$. Then using these facts, we can state in Eq.

(S18) that if $i \in \mathcal{P}$, then $\sum_j \Delta x_{ji} = 0$; and if $i \in \mathcal{N}$, then $\sum_j \Delta x_{ij} = 0$. Then, we can say that, only one of the terms is active in Eq. (S18) for positives and negatives:

$$\frac{\partial \mathcal{L}}{\partial s_i} = \underbrace{\frac{1}{Z} \sum_j \Delta x_{ij}}_{\text{Active if } i \in \mathcal{P}} - \underbrace{\frac{1}{Z} \sum_j \Delta x_{ji}}_{\text{Active if } i \in \mathcal{N}}. \tag{S19}$$

Considering that the value of a primary term cannot be less than its target, we have $\Delta x_{ij} \leq 0$, which implies $\frac{\partial \mathcal{L}}{\partial s_i} \leq 0$. So, we can take the absolute value outside of summation:

$$\sum_{i \in \mathcal{P}} \left| \frac{\partial \mathcal{L}}{\partial s_i} \right| = \left| \sum_{i \in \mathcal{P}} \frac{\partial \mathcal{L}}{\partial s_i} \right|, \tag{S20}$$

and using the fact identified in Eq. (S19) (i.e. for $i \in \mathcal{P}$, $\frac{\partial \mathcal{L}}{\partial s_i} = \frac{1}{Z} \sum_{j \in \mathcal{N}} \Delta x_{ij}$):

$$\left| \sum_{i \in \mathcal{P}} \frac{1}{Z} \sum_{j \in \mathcal{N}} \Delta x_{ij} \right| = \left| \frac{1}{Z} \sum_{i \in \mathcal{P}} \sum_{j \in \mathcal{N}} \Delta x_{ij} \right|. \tag{S21}$$

Simply interchanging the indices and the order of summations, and then reorganizing the constant $\frac{1}{Z}$ respectively yields:

$$\left| \frac{1}{Z} \sum_{j \in \mathcal{P}} \sum_{i \in \mathcal{N}} \Delta x_{ji} \right| = \left| \frac{1}{Z} \sum_{i \in \mathcal{N}} \sum_{j \in \mathcal{P}} \Delta x_{ji} \right| = \left| \sum_{i \in \mathcal{N}} \frac{1}{Z} \sum_{j \in \mathcal{P}} \Delta x_{ji} \right|. \tag{S22}$$

Note that for $i \in \mathcal{N}$, $\frac{\partial \mathcal{L}}{\partial s_i} = -\frac{1}{Z} \sum_{j \in \mathcal{P}} \Delta x_{ji}$, and hence $\frac{1}{Z} \sum_{j \in \mathcal{P}} \Delta x_{ji} = -\frac{\partial \mathcal{L}}{\partial s_i}$. Replacing $\frac{1}{Z} \sum_{j \in \mathcal{P}} \Delta x_{ji}$;

$$\left| \sum_{i \in \mathcal{N}} -\frac{\partial \mathcal{L}}{\partial s_i} \right|. \tag{S23}$$

Since, for $i \in \mathcal{N}$, $\frac{\partial \mathcal{L}}{\partial s_i} = -\frac{1}{Z} \sum_{j \in \mathcal{P}} \Delta x_{ji}$ is greater or equal to zero, the proof follows:

$$\left| \sum_{i \in \mathcal{N}} -\frac{\partial \mathcal{L}}{\partial s_i} \right| = \left| \sum_{i \in \mathcal{N}} \frac{\partial \mathcal{L}}{\partial s_i} \right| = \sum_{i \in \mathcal{N}} \left| \frac{\partial \mathcal{L}}{\partial s_i} \right|. \tag{S24}$$

$\square$

## S4 Normalized Discounted Cumulative Gain (NDCG) Loss and Its Gradients: Another Case Example for our Generalized Framework

In the following we define and derive the gradients of the NDCG Loss [11] following our generalized framework presented in Section 3 of our main paper.

The NDCG loss is defined as:

$$\mathcal{L}^{\text{NDCG}} = 1 - \frac{1}{G_{max}} \sum_{i \in \mathcal{P}} G(i) = \frac{G_{max} - \sum_{i \in \mathcal{P}} G(i)}{G_{max}} = \sum_{i \in \mathcal{P}} \frac{G_{max}/|\mathcal{P}| - G(i)}{G_{max}}. \tag{S25}$$

Note that different from AP Loss and aLRP Loss, here $Z$ turns out to be 1, which makes sense since NDCG is normalized by definition. Also, based on Eq. S25, one can identify NDCG Error on a

positive as: $\ell^{\mathrm{NDCG}}(i) = \frac{G_{max}/|\mathcal{P}|-G(i)}{G_{max}}$ such that $G(i) = \frac{1}{\log_2(1+\mathrm{rank}(i))}$ and $G_{max} = \sum\limits_{i=1}^{|\mathcal{P}|} \log_2(1+i)$.

Similar to AP and aLRP Loss, using $p(j|i) = \frac{H(x_{ij})}{N_{FP}(i)}$, the primary term of the NDCG Loss is $L_{ij}^{\mathrm{NDCG}} = \ell^{\mathrm{NDCG}}(i)p(j|i)$ (line 1 of Algorithm 1 in the paper). When the positive example $i$ is ranked properly, $G(i) = \frac{1}{\log_2(1+1)} = 1$, and resulting desired NDCG Error is (line 2 of Algorithm 1):

$$\ell^{\mathrm{NDCG}}(i)^* = \frac{G_{max}/|\mathcal{P}| - 1}{G_{max}}, \tag{S26}$$

yielding a target primary term $L_{ij}^{\mathrm{NDCG}*} = \ell_i^{\mathrm{NDCG}*}p(j|i)$. Using $L_{ij}^{\mathrm{NDCG}}$ and $L_{ij}^{\mathrm{NDCG}*}$, the update can be calculated as follows (line 3 of Algorithm 1):

$$\Delta x_{ij} = L_{ij}^{\mathrm{NDCG}*} - L_{ij}^{\mathrm{NDCG}} = \left( \ell^{\mathrm{NDCG}}(i)^* - \ell^{\mathrm{NDCG}}(i) \right) p(j|i), \tag{S27}$$

$$= \left( \frac{G_{max}/|\mathcal{P}| - G(i)}{G_{max}} - \frac{G_{max}/|\mathcal{P}| - 1}{G_{max}} \right) \frac{H(x_{ij})}{N_{FP}(i)}, \tag{S28}$$

$$= \frac{1 - G(i)}{G_{max}} \frac{H(x_{ij})}{N_{FP}(i)}, \tag{S29}$$

and one can compute the gradients using Eq. 5 in the paper (line 4 of Algorithm 1).

## S5    Computing aLRP Loss and its Gradients

This section presents the algorithm to compute aLRP Loss in detail along with an analysis of space and time complexity. For better understanding, bold font denotes multi-dimensional data structures (which can be implemented by vectors, matrices or tensors). Algorithm S1 describes the steps to compute aLRP Loss along with the gradients for a given mini-batch.

**Description of the inputs**: $\mathbf{S}$ is the raw output of the classification branch, namely logits. For localisation, as done by IoU-based localisation losses [18, 16], the raw localisation outputs need to be converted to the boxes, which are denoted by $\mathbf{B}$. We assume that $\mathbf{M}$ stores $-1$ for ignored anchors and $0$ for negative anchors. For positive anchors, $\mathbf{M}$ stores the index of the ground truth (i.e. $\{1, ..., |\hat{\mathbf{B}}|\}$, where $\hat{\mathbf{B}}$ is a list of ground boxes for the mini-batch). Hence, we can find the corresponding ground truth for a positive anchor only by using $\mathbf{M}$. $\delta$ is the smoothness of the piecewise linear function defined in Eq. S30 and set to 1 following AP Loss. We use the self-balance ratio, $\frac{\mathcal{L}^{\mathrm{aLRP}}}{\mathcal{L}_{cls}^{\mathrm{aLRP}}}$, by averaging over its values from the previous epoch. We initialize it as 50 (i.e. see Table 4 in the paper).

**Part 1: Initializing Variables**: Lines 2-10 aim to initialize the necessary data from the inputs. While this part is obvious, please note that line 8 determines a threshold to select the relevant negative outputs. This is simply due to Eq. S30 and the gradients of these negative examples with scores under this threshold are zero. Therefore, for the sake of time and space efficiency, they are ignored.

**Part 2: Computing Unnormalized Localisation Errors**: Lines 12-14 compute unnormalized localisation error on each positive example. Line 12 simply finds the localisation error of each positive example and line 13 sorts these errors with respect to their scores in descending order, and Line 14 computes the cumulative sum of the sorted errors with cumsum function. In such a way, the example with the larger scores contributes to the error computed for each positive anchor with smaller scores. Note that while computing the nominator of the $\mathcal{L}_{loc}^{\mathrm{aLRP}}$, we employ the step function (not the piecewise linear function), since we can safely use backpropagation.

**Part 3: Computing Gradient and Error Contribution from Each Positive**: Lines 16-32 compute the gradient and error contribution from each positive example. To do so, Line 16 initializes necessary data structures. Among these data structures, while $\mathcal{L}_{\mathbf{loc}}^{\mathrm{LRP}}$, $\mathcal{L}_{\mathbf{cls}}^{\mathrm{LRP}}$ and $\frac{\partial \mathcal{L}^{\mathrm{aLRP}}}{\partial \mathbf{S}_+}$ are all with size $|\mathcal{P}|$, $\frac{\partial \mathcal{L}^{\mathrm{aLRP}}}{\partial \mathbf{S}_-}$ has size $|\hat{\mathcal{N}}|$, where $\hat{\mathcal{N}}$ is the number of negative examples after ignoring the ones with scores less than $\tau$ in Line 8, and obviously $|\hat{\mathcal{N}}| \leq |\mathcal{N}|$. The loop iterates over each positive example by computing LRP values and gradients since aLRP is defined as the average LRP values over positives

(see Eq. 9 in the paper). Lines 18-22 computes the relation between the corresponding positive with positives and relevant negatives, each of which requires the difference transformation followed by piecewise linear function:

$$H(x) = \begin{cases} 0, & x < -\delta \\ \frac{x}{2\delta} + 0.5, & -\delta \le x \le \delta \\ 1, & \delta < x. \end{cases} \tag{S30}$$

Then, using these relations, lines 23-25 compute the rank of the $i$th examples within positive examples, number of negative examples with larger scores (i.e. false positives) and rank of the example. Lines 26 and 27 compute aLRP classification and localisation errors on the corresponding positive example. Note that to have a consistent denominator for total aLRP, we use rank to normalize both of the components. Lines 28-30 compute the gradients. While the local error is enough to determine the unnormalized gradient of a positive example, the gradient of a negative example is accumulated through the loop.

**Part 4: Computing aLRP Loss and Gradients**: Lines 34-40 simply derive the final aLRP value by averaging over LRP values (lines 34-36), normalize the gradients (lines 37-38) and compute gradients wrt the boxes (line 39) and applies self balancing (line 40).

### S5.1 Time Complexity

- First 16 lines of Algorithm S1 require time between $\mathcal{O}(|\mathcal{P}|)$ and $\mathcal{O}(|\mathcal{N}|)$. Since for the object detection problem, the number of negative examples is quite larger than number of positive anchors (i.e. $|\mathcal{P}| << |\mathcal{N}|$), we can conclude that the time complexity of first 13 lines is $\mathcal{O}(|\mathcal{N}|)$.

- The bottleneck of the algorithm is the loop on lines 17-32. The loop iterates over each positive example, and in each iteration while lines 21, 24 and 30 are executed for relevant negative examples, the rest of the lines is executed for positive examples. Hence the number of operations for each iteration is $\max(|\mathcal{P}|, |\hat{\mathcal{N}}|)$ (i.e. number of relevant negatives, see lines 8-9), and overall these lines require $\mathcal{O}(|\mathcal{P}| \times \max(|\mathcal{P}|, |\hat{\mathcal{N}}|))$. Note that, while in the early training epochs, $|\hat{\mathcal{N}}| \approx |\mathcal{N}|$, as the training proceeds, the classifier tends to distinguish positive examples from negative examples very well, and $|\hat{\mathcal{N}}|$ significantly decreases implying faster mini-batch iterations.

- The remaining lines between 26-33 again require time between $\mathcal{O}(|\mathcal{P}|)$ and $\mathcal{O}(|\mathcal{N}|)$.

Hence, we conclude that the time complexity of Algorithm S1 is $\mathcal{O}(|\mathcal{N}| + |\mathcal{P}| \times \max(|\mathcal{P}|, |\hat{\mathcal{N}}|))$.

Compared to AP Loss;

- aLRP Loss includes an extra computation of aLRP localisation component (i.e. lines 12-14, 27. Each of these lines requires $\mathcal{O}(|\mathcal{P}|)$).

- aLRP Loss includes an additional summation while computing the gradients with respect to the scores of the positive examples in line 29 requiring $\mathcal{O}(|\mathcal{P}|^2)$.

- aLRP Loss discards interpolation (i.e. using interpolated AP curve), which can take up to $\mathcal{O}(|\mathcal{P}| \times |\hat{\mathcal{N}}|)$.

### S5.2 Space Complexity

Algorithm S1 does not require any data structure larger than network outputs (i.e. $\mathbf{B}$, $\mathbf{S}$). Then, we can safely conclude that the space complexity is similar to all of the common loss functions that is $\mathcal{O}(|\mathbf{S}|)$.

## S6 Details of aLRP Loss

This section provides details for aLRP Loss.

**Algorithm S1** The algorithm to compute aLRP Loss for a mini-batch.

---

**Input: S**: Logit predictions of the classifier for each anchor,
  **B**: Box predictions of the localization branch from each anchor,
  $\hat{\mathbf{B}}$: Ground truth (GT) boxes,
  **M**: Matching of the Anchors with the GTs Boxes.
  $\delta$: Smoothness of the piecewise linear function ($\delta = 1$ by default).
  $w_{ASB}$: ASB weight, computed using $\frac{\mathcal{L}^{\text{aLRP}}}{\mathcal{L}^{\text{aLRP}}_{cls}}$ values from previous epoch.

**Output:** $\mathcal{L}^{\text{aLRP}}$: aLRP loss, $\frac{\partial \mathcal{L}^{\text{aLRP}}}{\partial \mathbf{S}}$: Gradients wrt logits, $\frac{\partial \mathcal{L}^{\text{aLRP}}}{\partial \mathbf{B}}$: Gradients wrt boxes.

1: // ==== PART 1: Initializing Variables ====
2: $\mathbf{idx}_+ :=$ The indices of $\mathbf{M}$ where $\mathbf{M} > 0$.
3: $\mathbf{M}_+ :=$ The values of $\mathbf{M}$ where $\mathbf{M} > 0$.
4: $\mathbf{B}_+ :=$ The values of $\mathbf{B}$ at indices $\mathbf{idx}_+$.
5: $\mathbf{S}_+ :=$ The values of $\mathbf{S}$ at indices $\mathbf{idx}_+$.
6: $\mathbf{idx}^{\mathbf{sorted}}_+ :=$ The indices of $\mathbf{S}_+$ once it is sorted in descending order.
7: $\mathbf{S}^{\mathbf{sorted}}_+ :=$ The values of $\mathbf{S}_+$ when ordered according to $\mathbf{idx}^{\mathbf{sorted}}_+$.
8: $\tau = \min(\mathbf{S}_+) - \delta$.
9: $\mathbf{idx}_- :=$ The indices of $\mathbf{M}$ where $\mathbf{M} = 0$ and $s_j \geq \tau$ (i.e. relevant negatives only).
10: $\mathbf{S}_- :=$ The values of $\mathbf{S}$ at indices $\mathbf{idx}_-$.
11: // ==== PART 2: Computing Unnormalized Localisation Errors ====
12: $\mathcal{E}_{\mathbf{Loc}} = \frac{1 - \text{IoU}(\mathbf{B}_+, \hat{\mathbf{B}}_+)}{1-\tau}$. (or $\mathcal{E}_{\mathbf{Loc}} = \frac{(1 - \text{GIoU}(\mathbf{B}_+, \hat{\mathbf{B}}_+))/2}{1-\tau}$ for GIoU Loss [16].)
13: $\mathcal{E}^{\mathbf{sorted}}_{\mathbf{Loc}} :=$ The values of $\mathcal{E}_{\mathbf{Loc}}$ when ordered according to $\mathbf{idx}^{\mathbf{sorted}}_+$.
14: $\mathcal{E}^{\mathbf{cumsum}}_{\mathbf{Loc}} = \text{cumsum}(\mathcal{E}^{\mathbf{sorted}}_{\mathbf{Loc}})$
15: // ==== PART 3: Computing Gradient and Error Contribution from Each Positive ====
16: Initialize , $\mathcal{L}^{\text{LRP}}_{\mathbf{loc}}$, $\mathcal{L}^{\text{LRP}}_{\mathbf{cls}}$, $\frac{\partial \mathcal{L}^{\text{aLRP}}}{\partial \mathbf{S}_+}$ and $\frac{\partial \mathcal{L}^{\text{aLRP}}}{\partial \mathbf{S}_-}$.
17: **for each** $s_i \in \mathbf{S}^{\mathbf{sorted}}_+$ **do**
18: $\quad \mathbf{X}_+ :=$ Difference transform of $s_i$ with the logit of each positive example.
19: $\quad \mathbf{R}_+ :=$ The relation of $i \in \mathcal{P}$ with each $j \in \mathcal{P}$ using Eq. S30 with input $\mathbf{X}_+$.
20: $\quad \mathbf{R}_+[i] = 0$
21: $\quad \mathbf{X}_- :=$ Difference transform of $s_i$ with the logit of each negative example.
22: $\quad \mathbf{R}_- :=$ The relation of $i \in \mathcal{P}$ with each $j \in \mathcal{N}$ using Eq. S30 with input $\mathbf{X}_+$.
23: $\quad \text{rank}_+ = 1 + \text{sum}(\mathbf{R}_+)$
24: $\quad \text{FP} = \text{sum}(\mathbf{R}_-)$
25: $\quad \text{rank} = \text{rank}_+ + \text{FP}$
26: $\quad \mathcal{L}^{\text{LRP}}_{\mathbf{cls}}[i] = \text{FP}/\text{rank}$
27: $\quad \mathcal{L}^{\text{LRP}}_{\mathbf{loc}}[i] = \mathcal{E}^{\mathbf{cumsum}}_{\mathbf{Loc}}[i]/\text{rank}$
28: $\quad$ **if** $\text{FP} \geq \epsilon$ **then** //For stability set $\epsilon$ to a small value (e.g. $1e-5$)

29: $$\frac{\partial \mathcal{L}^{\text{aLRP}}}{\partial \mathbf{S}_+}[i] = -\left( \text{FP} + \sum_{i \in P} \mathbf{R}_+[i] \times \mathcal{E}^{cumsum}_{Loc}[i] \right)/\text{rank}$$

30: $$\frac{\partial \mathcal{L}^{\text{aLRP}}}{\partial \mathbf{S}_-} += \left( -\frac{\partial \mathcal{L}^{\text{aLRP}}}{\partial \mathbf{S}_+}[i] \times \frac{\mathbf{R}_-}{\text{FP}} \right)$$

31: $\quad$ **end if**
32: **end for**
33: // ==== PART 4: Computing the aLRP Loss and Gradients ====
34: $\mathcal{L}^{\text{aLRP}}_{cls} = \text{mean}(\mathcal{L}^{\text{LRP}}_{\mathbf{cls}})$
35: $\mathcal{L}^{\text{aLRP}}_{loc} = \text{mean}(\mathcal{L}^{\text{LRP}}_{\mathbf{loc}})$
36: $\mathcal{L}^{\text{aLRP}} = \mathcal{L}^{\text{aLRP}}_{cls} + \mathcal{L}^{\text{aLRP}}_{loc}$
37: Place $\frac{\partial \mathcal{L}^{\text{aLRP}}}{\partial \mathbf{S}_+}$ and $\frac{\partial \mathcal{L}^{\text{aLRP}}}{\partial \mathbf{S}_-}$ into $\frac{\partial \mathcal{L}^{\text{aLRP}}}{\partial \mathbf{S}}$ also by setting the gradients of remaining examples to 0.
38: $\frac{\partial \mathcal{L}^{\text{aLRP}}}{\partial \mathbf{S}} /= |\mathcal{P}|$
39: Compute $\frac{\partial \mathcal{L}^{\text{aLRP}}}{\partial \mathbf{B}}$ (possibly using autograd property of a deep learning library or refer to the supp. mat. of [16] for the gradients of GIoU and IoU Losses.
40: $\frac{\partial \mathcal{L}^{\text{aLRP}}_{loc}}{\partial \mathbf{B}} \times= w_{ASB}$
41: **return** $\frac{\partial \mathcal{L}^{\text{aLRP}}}{\partial \mathbf{S}}$, $\frac{\partial \mathcal{L}^{\text{aLRP}}}{\partial \mathbf{B}}$ and $\mathcal{L}^{\text{aLRP}}$.

---

### S6.1 A Soft Sampling Perspective for aLRP Localisation Component

In sampling methods, the contribution $(w_i)$ of the $i$th bounding box to the loss function is adjusted as follows:

$$\mathcal{L} = \sum_{i \in \mathcal{P} \cup \mathcal{N}} w_i \mathcal{L}(i), \tag{S31}$$

where $\mathcal{L}(i)$ is the loss of the $i$th example. Hard and soft sampling approaches differ on the possible values of $w_i$. For the hard sampling approaches, $w_i \in \{0, 1\}$, thus a BB is either selected or discarded. For soft sampling approaches, $w_i \in [0, 1]$, i.e. the contribution of a sample is adjusted with a weight and each BB is somehow included in training. While this perspective is quite common to train the classification branch [1, 8]; the localisation branch is conventionally trained by hard sampling with some exceptions (e.g. CARL [1] sets $w_i = s_i$ where $s_i$ is the classification score).

Here, we show that, in fact, what aLRP localisation component does is soft sampling. To see this, first let us recall the definition of the localisation component:

$$\mathcal{L}_{loc}^{\text{aLRP}} = \frac{1}{|\mathcal{P}|} \sum_{i \in \mathcal{P}} \frac{1}{\text{rank}(i)} \left( \mathcal{E}_{loc}(i) + \sum_{k \in \mathcal{P}, k \neq i} \mathcal{E}_{loc}(k) H(x_{ik}) \right), \tag{S32}$$

which is differentiable with respect to the box parameters as discussed in the paper. With a ranking-based formulation, note that (i) the localisation error of a positive example $i$ (i.e. $\mathcal{E}_{loc}(i)$) contributes each LRP value computed on a positive example $j$ where $s_i \geq s_j$ (also see Fig. 2 in the paper), and (ii) each LRP value computed on a positive example $i$ is normalized by $\text{rank}(i)$. Then, setting $\mathcal{L}(i) = \mathcal{E}_{loc}(i)$ in Eq. S31 and accordingly taking Eq. S32 in $\mathcal{E}_{loc}(i)$ paranthesis, the weights of the positive examples (i.e. $w_i = 0$ for negatives for the localisation component) are:

$$w_i = \frac{1}{|\mathcal{P}|} \left( \left( \sum_{k \in \mathcal{P}, k \neq i} \frac{H(x_{ki})}{\text{rank}(k)} \right) + \frac{1}{\text{rank}(i)} \right). \tag{S33}$$

Note that $\mathcal{L}(i)$ is based on a differentiable IoU-based regression loss and $w_i$ is its weight, which is a scaler. As a result $H(x_{ki})$ in Eq. S33 does not need to be smoothed and we use a unit-step function (see line 14 in Algorithm S1).

### S6.2 The Relation between aLRP Loss Value and Total Gradient Magnitudes

Here, we identify the relation between the loss value and the total magnitudes of the gradients following the generalized framework due to the fact that it is a basis for our self-balancing strategy introduced in Section 4.2 as follows:

$$\sum_{i \in \mathcal{P}} \left| \frac{\partial \mathcal{L}}{\partial s_i} \right| = \sum_{i \in \mathcal{N}} \left| \frac{\partial \mathcal{L}}{\partial s_i} \right| \approx \mathcal{L}^{\text{aLRP}}. \tag{S34}$$

Since we showed in Section S3 that $\sum_{i \in \mathcal{P}} \left| \frac{\partial \mathcal{L}}{\partial s_i} \right| = \sum_{i \in \mathcal{N}} \left| \frac{\partial \mathcal{L}}{\partial s_i} \right|$, here we show that the loss value is approximated by the total magnitude of gradients. Recall from Eq. (S21) that total gradients of the positives can be expressed as:

$$\sum_{i \in \mathcal{P}} \left| \frac{\partial \mathcal{L}}{\partial s_i} \right| = \left| \frac{1}{|\mathcal{P}|} \sum_{i \in \mathcal{P}} \sum_{j \in \mathcal{N}} \Delta x_{ij} \right|. \tag{S35}$$

Since $\Delta x_{ij} \leq 0$, we can discard the absolute value by multiplying it by $-1$:

$$-\frac{1}{|\mathcal{P}|} \sum_{i \in \mathcal{P}} \sum_{j \in \mathcal{N}} \Delta x_{ij}. \tag{S36}$$

Replacing the definition of the $\Delta x_{ij}$ by $L_{ij}^* - L_{ij}$ yields:

$$-\frac{1}{|\mathcal{P}|}\sum_{i\in\mathcal{P}}\sum_{j\in\mathcal{N}}(L_{ij}^* - L_{ij}) = -\frac{1}{|\mathcal{P}|}\left(\sum_{i\in\mathcal{P}}\sum_{j\in\mathcal{N}}L_{ij}^* - \sum_{i\in\mathcal{P}}\sum_{j\in\mathcal{N}}L_{ij}\right) \quad \text{(S37)}$$

$$= \frac{1}{|\mathcal{P}|}\sum_{i\in\mathcal{P}}\sum_{j\in\mathcal{N}}L_{ij} - \frac{1}{|\mathcal{P}|}\sum_{i\in\mathcal{P}}\sum_{j\in\mathcal{N}}L_{ij}^*. \quad \text{(S38)}$$

Using Theorem 1, the first part (i.e $\frac{1}{|\mathcal{P}|}\sum_{i\in\mathcal{P}}\sum_{j\in\mathcal{N}}L_{ij}$) yields the loss value, $\mathcal{L}$. Hence:

$$\sum_{i\in\mathcal{P}}\left|\frac{\partial\mathcal{L}}{\partial s_i}\right| = \mathcal{L} - \frac{1}{|\mathcal{P}|}\sum_{i\in\mathcal{P}}\sum_{j\in\mathcal{N}}L_{ij}^*. \quad \text{(S39)}$$

Reorganizing the terms, the difference between the total gradients of positives (or negatives, since they are equal – see Theorem 2) and the loss values itself is the sum of the targets normalized by number of positives:

$$\mathcal{L} - \sum_{i\in\mathcal{P}}\left|\frac{\partial\mathcal{L}}{\partial s_i}\right| = \frac{1}{|\mathcal{P}|}\sum_{i\in\mathcal{P}}\sum_{j\in\mathcal{N}}L_{ij}^*. \quad \text{(S40)}$$

Compared to the primary terms, the targets are very small values (if not 0). For example, for AP Loss $L_{ij}^{\mathrm{AP}*} = 0$, and hence, loss is equal to the sum of the gradients: $\mathcal{L} = \sum_{i\in\mathcal{P}}\left|\frac{\partial\mathcal{L}}{\partial s_i}\right|$.

As for aLRP Loss, the target of a primary term is $\frac{\mathcal{E}_{loc}(i)}{\mathrm{rank}(i)}\frac{H(x_{ij})}{N_{FP}(i)}$, hence if $H(x_{ij}) = 0$, then the target is also 0. Else if $H(x_{ij}) = 1$, then it implies that there are some negative examples with larger scores, and $\mathrm{rank}(i)$ and $N_{FP}(i)$ are getting larger depending on these number of negative examples, which causes the denominator to grow, and hence yielding a small target as well. Then ignoring this term, we conclude that:

$$\sum_{i\in\mathcal{P}}\left|\frac{\partial\mathcal{L}}{\partial s_i}\right| = \sum_{i\in\mathcal{N}}\left|\frac{\partial\mathcal{L}}{\partial s_i}\right| \approx \mathcal{L}^{\mathrm{aLRP}}. \quad \text{(S41)}$$

### S6.3 Self-balancing the Gradients Instead of the Loss Value

Instead of localisation the loss, $\mathcal{L}_{loc}^{\mathrm{aLRP}}$, we multiply $\partial\mathcal{L}/\partial B$ by the average $\mathcal{L}^{\mathrm{aLRP}}/\mathcal{L}_{loc}^{\mathrm{aLRP}}$ of the previous epoch. This is because formulating aLRP Loss as $\mathcal{L}_{loc}^{\mathrm{aLRP}} + w_r\mathcal{L}_{loc}^{\mathrm{aLRP}}$ where $w_r$ is a weight to balance the tasks is different from weighing the gradients with respect to the localisation output, $B$, since weighting the loss value (i.e. $\mathcal{L}_{loc}^{\mathrm{aLRP}} + w_r\mathcal{L}_{loc}^{\mathrm{aLRP}}$) changes the gradients of aLRP Loss with respect to the classification output as well since $\mathcal{L}_{loc}^{\mathrm{aLRP}}$, now weighed by $w_r$, is also ranking-based (has $\mathrm{rank}(i)$ term - see Eq. 11 in the paper). Therefore, we directly add the self balance term as a multiplier of $\partial\mathcal{L}/\partial B$ and backpropagate accordingly. On the other hand, from a practical perspective, this can simply be implemented by weighing the loss value, $\mathcal{L}_{loc}^{\mathrm{aLRP}}$ without modifying the gradient formulation for $\mathcal{L}_{cls}^{\mathrm{aLRP}}$.

## S7 Additional Experiments

This section presents more ablation experiments, the anchor configuration we use in our models and the effect of using a wrong target for the primary term in the error-driven update rule.

### S7.1 More Ablation Experiments: Using Self Balance and GIoU with AP Loss

We also test the effect of GIoU and our Self-balance approach on AP Loss, and present the results in Table S4:

- Using IoU-based losses with AP Loss improves the performance up to 1.0 AP as well and reaches 36.5 AP with GIoU loss.

Table S4: Using Self Balance and GIoU with AP Loss. For optimal LRP (oLRP), lower is better.

| $\mathcal{L}_c$ | $\mathcal{L}_r$ | SB | AP | $AP_{50}$ | $AP_{75}$ | $AP_{90}$ | oLRP | $\rho$ |
|---|---|---|---|---|---|---|---|---|
| AP Loss [2] | Smooth L1 | | 35.5 | 58.0 | 37.0 | 9.0 | 71.0 | 0.45 |
| | Smooth L1 | ✓ | 36.7 | 58.2 | 39.0 | 10.8 | 70.2 | 0.44 |
| | IoU Loss | | 36.3 | 57.9 | 37.9 | 11.8 | 70.4 | 0.44 |
| | IoU Loss | ✓ | 37.2 | 58.1 | 39.2 | 13.1 | 69.6 | 0.44 |
| | GIoU Loss | | 36.5 | 58.1 | 38.1 | 11.9 | 70.2 | 0.45 |
| | GIoU Loss | ✓ | 37.2 | 58.3 | 39.0 | 13.4 | 69.7 | 0.44 |
| aLRP Loss | with IoU | | 36.9 | 57.7 | 38.4 | 13.9 | 69.9 | 0.49 |
| | with IoU | ✓ | 38.7 | 58.1 | 40.6 | 17.4 | 68.5 | 0.48 |
| | with GIoU | ✓ | 38.9 | 58.5 | 40.5 | 17.4 | 68.4 | 0.48 |

- Our SB approach also improves AP Loss between 0.7 - 1.2 AP, resulting in 37.2AP as the best performing model without using $w_r$. However, it may not be inferred that SB performs better than constant weighting for AP Loss without a more thorough tuning of AP Loss since SB is devised to balance the gradients of localisation and classification outputs for aLRP Loss (see Section S6.2).

- Comparing with the best performing model of AP Loss with 37.2AP, (i) aLRP Loss has a 1.7AP and 1.3oLRP points better performance, (ii) the gap is 4.0AP for $AP_{90}$, and (iii) the correlation coeffient of aLRP Loss, preserves the same gap (0.48 vs 0.44 comparing the best models for AP and aLRP Losses), since applying these improvements (IoU-based losses and SB) to AP Loss does not have an effect on unifying branches.

## S7.2 Anchor Configuration

The number of anchors has a notable affect on the efficiency of training due to the time and space complexity of optimizing ranking-based loss functions by combining error-driven update and back-propagation. For this reason, different from original RetinaNet using three aspect ratios (i.e. $[0.5, 1, 2]$) and three scales (i.e. $[2^{0/2}, 2^{1/2}, 2^{2/2}]$) on each location, Chen et al. [2] preferred the same three aspect ratios, but reduced the scales to two as $[2^{0/2}, 2^{1/2}]$ to increase the efficiency of AP Loss. In our ablation experiments, except the one that we used ATSS [19], we also followed the same anchor configuration of Chen et al. [2].

One main contribution of ATSS is to simplify the anchor design by reducing the number of required anchors to a single scale and aspect ratio (i.e. ATSS uses 1/9 and 1/6 of the anchors of RetinaNet [8] and AP Loss [2] respectively), which is a perfect fit for our optimization strategy. For this reason, we used ATSS, however, we observed that the configuration in the original ATSS with a single aspect ratio and scale does not yield the best result for aLRP Loss, which may be related to the ranking nature of aLRP Loss which favors more examples to impose a more accurate ranking, loss and gradient computation. Therefore, different from ATSS configuration, we find it useful to set anchor scales $[2^{0/2}, 2^{1/2}]$ and $[2^{0/2}, 2^{1/2}, 2^{2/2}]$ for aLRPLoss500 and aLRPLoss800 respectively and use a single aspect ratio with 1 following the original design of ATSS.

## S7.3 Using a Wrong Target for the Primary Term in the Error-driven Update Rule

As discussed in our paper (Section 4.1, Equation 13), $L_{ij}^*$, the target value of the primary term $L_{ij}$ is non-zero due to the localisation error. It is easy to overlook this fact and assume that the target is zero. Fig. S3 presents this case where $L_{ij}^*$ is set to 0 (i.e. minimum value of aLRP). In such a case, the training continues properly, similar to that of the correct case, up to a point and then diverges. Note that this occurs when the positives start to be ranked properly but are still assigned gradients since $L_{ij}^* - L_{ij} \neq 0$ due to the nonzero localisation error. This causes $\sum_{i \in \mathcal{P}} \left| \frac{\partial \mathcal{L}}{\partial s_i} \right| > \sum_{i \in \mathcal{N}} \left| \frac{\partial \mathcal{L}}{\partial s_i} \right|$, violating Theorem 2 (compare min-rate and max-rate in Fig. S3). Therefore, assigning proper targets as indicated in Section 3 in the paper is crucial for balanced training.

Figure S3: **(left)** The rate of the total gradient magnitudes of negatives to positives. **(right)** Loss values.

### S7.4   Implementation Details for FoveaBox and Faster R-CNN

In this section, we provide more implementation details on the FoveaBox and Faster R-CNN models that we trained with different loss functions. All the models in this section are tested on COCO *minival*.

**Implementation Details of FoveaBox:** We train the models for 100 epochs with a learning rate decay at epochs 75 and 95. For aLRP Loss and AP Loss, we preserve the same learning rates used for RetinaNet (i.e. $0.008$ and $0.002$ for aLRP Loss and AP Loss respectively). As for the Focal Loss, we set the initial learning rate to $0.02$ following the linear scheduling hypothesis [13] (i.e. Kong et al. [7] set learning rate to $0.01$ and use a batch size of 16). Following AP Loss official implementation, the gradients of the regression loss (i.e. Smooth L1) are averaged over the output parameters of positive boxes for AP Loss. As for Focal Loss, we follow the mmdetection implementation which averages the total regression loss by the number of positive examples. The models are tested on COCO *minival* by preserving the standard design by mmdetection framework.

**Implementation Details of Faster R-CNN:** To train Faster R-CNN, we first replace the softmax classifier of Fast R-CNN by the class-wise sigmoid classifiers. Instead of heuristic sampling rules, we use all anchors to train RPN and top-1000 scoring proposals per image obtained from RPN to train Fast R-CNN (i.e. same with the default training except for discarding sampling). Note that, with aLRP Loss, the loss function consists of two independent losses instead of four in the original pipeline, hence instead of three scalar weights, aLRP Loss requires a single weight for RPN head, which we tuned as $0.20$. Following the positive-negative assignment rule of RPN, different from all the experiments, which use $\tau = 0.50$, $\tau = 0.70$ for aLRP Loss of RPN. We set the initial learning rate to $0.04$ following the linear scheduling hypothesis [13] for the baselines, and decreased by a factor of $0.10$ at epochs 75 and 95. Localisation loss weight is kept as 1 for L1 Loss and to 10 for GIoU Loss [3, 16]. The models are tested on COCO *minival* by preserving the standard design by mmdetection framework. We do not train Faster R-CNN with AP Loss due to the difficulty to tune Faster R-CNN for a different loss function.