[Reviews · NeurIPS 2020]

Review 1

Summary and Contributions: Follow the APloss which extends the Average Precision(AP) to the ranking-based loss function. This paper extends the Localization-Recall-Precision(LRP) to be ranking-aware. Benefit from LRP, aLRP simultaneously ranks based on the quality of classification and localization rather than single classification probability. Results show significant improvements over the APloss, and keeping the simplicity for tuning.

Strengths: + The proposed average Localization-Recall-Precision (aLRP) loss is well motivated. Turning the Localization-Recall-Precision(LRP) into a ranking-based loss is a clear contribution. + It seems aLRP loss can be integrated into many object detection networks with ease. + Significant improvements over the baseline of AP Loss[7] have been witnessed.

Weaknesses: + The experiment is only done for one method. It would be great to have more consistently good results on other stronger detectors, like “Fully Convolutional One-Stage Object Detection”. + Many recent works make great progress in detection label assignments , e.g.“Bridging the Gap Between Anchor-based and Anchor-free Detection via Adaptive Training Sample Selection”, which has surpassed the plain APloss by a large-margin. I wonder how these label-assign strategies work with the aLRP loss. Can they be further improved? + Using confidences of both the classification and localization to present the unified confidence for each anchor should have been modeling in FreeAnchor.

Correctness: Yes

Clarity: Yes

Relation to Prior Work: yes

Reproducibility: Yes

Additional Feedback:


Review 2

Summary and Contributions: This submission involves the average Localization-Recall-Precision (aLRP), a ranking-based loss function for classification and localisation in object detection. As a new ranking loss, aLRP is claimed to enforce high-quality localisation for high-precision classification.

Strengths: aLRP reduces multiple hyperparameters in a single-state detector to a single hyperparameter and thus simplifies the training of detectors. Plausible justification about the loss function is given in the supplementary material.

Weaknesses: The performance of object detection at 512 \times 512 resolution is lower than, if not comparable to, many state-of-the-art detectors. Simplying multiple parameters to a single parameter is amazing. It will be more interesting if the performance can outperform the state-of-the-arts. Furthermfore, the general applicability of aLRP to object detectors, like FasterRCNN, SSD, or YOLO, was not valided. I am also wondering can the aLPR approach works given test images of very sparse objects.

Correctness: Probably correct.

Clarity: The main idea of this paper is conveyed. But this is a room to improve the equation and justification. For example, the `rank(i)" in many equations can confuse the readers.

Relation to Prior Work: The difference and relations to previous work are clearly discussed.

Reproducibility: Yes

Additional Feedback: Please refer to the weakness part.


Review 3

Summary and Contributions: This paper proposes an average Localization -Recall-Precision (aLRP) loss for object detection. The loss has the following advantages: (1) it can unify the classification and regression in object detection. (2) It can naturally enforce high-quality boxes for high classification scores. (3) It can bypass the imbalance between positive and negative samples. Experiments on COCO shows that the method can improve the performance of the baseline object detector by a large margin, without affecting the inference speed.

Strengths: The proposed loss function really has a lot of desired properties. It can solve a lot of probelems such as sample imbalance, the gap between classification and regression and etc.. I think it is a good paper. The proposed method also has only one hyper-parameter, which can largely reduce the effort of hyperparameter tuning. The proposed loss improves the performance of the baseline detector by a large margin.

Weaknesses: I can't find obvious flaws of the work. But I encouge the authors to try the loss on more detectors, which will make their claims more convincing. For example, can the loss function improve the performance of anchor-free detectors such as FCOS, or the two-stage Faster R-CNN?

Correctness: Yes.

Clarity: The paper is well-written.

Relation to Prior Work: Yes.

Reproducibility: Yes

Additional Feedback: The authors address my questions in rebuttal, so I keep my score.


Review 4

Summary and Contributions: The paper proposes a loss function (aLRP loss) that can be seen as an extension of ranking based AP loss for both classification and regression objectives in object detection. It promotes detections with larger IoU to have higher scores and suppresses the detections with a high score and low IoU. The method achieves promising results with only one hyper-parameter.

Strengths: The proposed loss function is novel and intuitive. Although joint improvement of classification and regression branch architectures are proposed in the literature, to the best of my knowledge, a loss function targeting both these objectives in an object detector is novel. The effectiveness of the proposed method is demonstrated through thorough ablation experiments and SOTA comparisons on COCO dataset. It is interesting to see that a simple loss function can provide ~5% performance gain on RetinaNet.

Weaknesses: 1. Line 124, both classification and regression losses are given equal weight ‘1’. An ablation study introducing an additional loss weight hyper-parameter or learned loss weight like [11] will be helpful. 2. Intuitively, having a localization term can help the classification objective since it is hard to define a clear boundary and say all boxes having “x” overlap with the ground-truth bounding box are objects. But it is not clear to me the intuition behind using the classifier branch output (rank (i) ) for the regression task. 3. The proposed method is demonstrated only on a single-stage RetinaNet object detector. I strongly advise showing the effectiveness of the proposed method on other object detectors. 4. Is the method applicable to two-stage object detectors and recent anchor-free methods like FCOS?. Experimental results will be helpful.

Correctness: Seems to be correct.

Clarity: The high-level motivation is clear, but need some effort to understand the paper.

Relation to Prior Work: Yes. The paper discusses its differences with related loss functions.

Reproducibility: Yes

Additional Feedback: After Rebuttal --------------- The rebuttal has addressed most of my concerns, so I will maintain my initial positive rating. As mentioned in the rebuttal, I suggest the authors to report the performances on (i) FoveaBox/FCOS, (ii) other single-stage, and (iii) two stage object detectors in the final draft.

[Author Response · NeurIPS 2020]

We thank all the reviewers for their valuable feedback. To summarize, we received three "Good Paper; accept" and a "Marginally above the acceptance threshold" ratings, which confirm the significance of our work for the community. For the rebuttal, we incorporated aLRP Loss into the "mmdetection framework", which includes the implementations of all major detectors, performed additional experiments with an anchor-free detector (FoveaBox [A]) and a label-assignment method (ATSS [B]), and started more experiments with other methods whose results will be included in the final paper.

**R1.1:** "*The experiment is only done for one method...*" **Authors:** We only tested aLRP Loss on RetinaNet following recent similar works (e.g. FreeAnchor [30], DR Loss [22], AP Loss [7]). Table A1 presents results with FoveaBox [A], a state-of-the-art anchor-free detector (similar to FCOS), by replacing Focal Loss and SmoothL1 with aLRP Loss. We observed that aLRP Loss (i) improves FoveaBox and (ii) discards four hyperparameters of the default loss function. However, these results are not final because we used RetinaNet's optimal learning rate and schedule, which we will tune for FoveaBox. Our experiments are still in progress for two-stage detectors and other one-stage detectors.

**R1.2:** "*...I wonder how these label-assign strategies work with the aLRP loss. Can they be further improved?*" **Authors:** Table A2 shows that aLRP Loss and ATSS [B] are complementary. aLRP Loss improves ATSS by around 1 AP. Furthermore, we notice that ATSS decreases training time of aLRP Loss by using fewer anchors (i.e. positives). We thank R1 for this contribution. We will provide a discussion on the combination of ATSS and aLRPLoss in the final version.

**R1.3:** "*... the unified confidence for each anchor should have been modeling in FreeAnchor.*" **Authors:** Agreed. We will move FreeAnchor under "Methods Combining Branches" in Table 4.

**R2.1:** "*The performance of object detection at $512 \times 512$ resolution is lower ... It will be more interesting if the performance can outperform the state-of-the-arts.*" **Authors:** aLRP Loss does outperform all methods with the same backbone and similar test scales. For example, for 500 scale, we improve the closest counterpart, HSD [2], by 1.5 AP. Please see lines 218-220 and Table 4.

**R2.2:** " *the general applicability of aLRP to object detectors ... was not valided.*" **Authors:** Please see **R1.1** and **R1.2**.

**R2.3:** "*... can the aLPR approach works given test images of very sparse objects.*" **Authors:** Table A3 compares the performance of methods on COCO minival for different number of objects per image. aLRPLoss has a significant gain (i.e. $\sim 4.5$ AP) for very sparse images (i.e. 0-3 interval), and outperforms other methods in each sparsity level.

**R3.1:** "*... I encouge the authors to try the loss on more detectors...*" **Authors:** Please see **R1.1** and **R1.2**.

**R4.1:** "*... An ablation study introducing an additional loss weight hyper-parameter or learned loss weight like [11] will be helpful.*" **Authors:** As we wrote in Section 4.2, we use a self-balancing (SB) strategy between localisation and classification, which is hyper-parameter free, simple and theoretically validated. Table A4 compares our SB method with scalar weighting. The results suggest that SB discards tuning $w_r$, and slightly yields better performance. We thank R4 for pointing out this and will include this ablation experiment in the paper.

**R4.2:** "*...not clear to me the intuition behind using the classifier branch output (rank (i) ) for the regression task.*" **Authors:** A large rank for a positive example implies a lower score ($s_i$), therefore including rank($i$) in the denominator decreases its contribution to the localisation loss, which focuses the training of the localisation branch on positive boxes with larger scores (i.e. those with smaller rank($i$)). Fig.2(c) and Lines 146-153 discuss this interaction. We will make it more clear. Also note that rank($i$) is the direct outcome of converting LRP to loss term $\ell^{\mathrm{LRP}}(i)$ and hence ensures comparable ranges for classification and localisation components.

**R4.3:** "*...I strongly advise showing the effectiveness of the proposed method on other object detectors. Is the method applicable to two-stage object detectors and recent anchor-free methods...*" **Authors:** Please see **R1.1** and **R1.2**.

[A] Kong et al., "FoveaBox: Beyond Anchor-based Object Detector", IEEE Transactions on Image Processing, 2020.
[B] Zhang et al., "Bridging the Gap Between Anchor-based and Anchor-free Detection...", CVPR, 2020.

Table A1: Using aLRP Loss with FoveaBox [A].

| Method | Epochs | | |
|---|---|---|---|
| | 50 | 75 | 100 |
| Focal L.+SL1 | 29.5 | 38.5 | 39.8 |
| aLRP L. | **34.0** | **39.7** | **40.3** |

Table A2: Using aLRP Loss with ATSS [B].

| Method | AP |
|---|---|
| ATSS | 30.9 |
| ATSS + aLRP L. | **32.0** |

Table A3: Effect of Sparsity.

| Method | # of objects/image | | | |
|---|---|---|---|---|
| | 0-3 | 4-10 | 11-20 | 21-62 |
| Focal L.+SL1 | 52.5 | 39.3 | 31.5 | 22.9 |
| AP L.+SL1 | 51.9 | 38.3 | 30.4 | 23.9 |
| aLRP L. | **56.3** | **42.0** | **33.5** | **25.5** |

Table A4: Ablation analysis for additional scalar localisation task weight, $w_r$. SB: Self-Balance (see Sec. 4.2)

| $w_r$ | 1 | 5 | 10 | 15 | SB |
|---|---|---|---|---|---|
| AP | 27.3 | 30.5 | 30.8 | 30.4 | **30.9** |

*The models in Tables A2 and A4 are trained on COCO minitrain (https://github.com/giddyyupp/coco-minitrain), approximately a quarter of the coco-trainval. All models use ResNet-50 backbone, follow the training details for 500 scale described in Section 5 and are tested on COCO minival.*

[Meta-Review · NeurIPS 2020]

This paper proposes an improved detection method by a combination of detection and ranking loss. It received a unanimous positive assessment from reviewers. It was felt that the combination of losses was novel and intuitive, though there were suggestions that reporting performance on additional detectors would make the work stronger.